# PDTrim: Targeted Pruning for Prefill-Decode Disaggregation in Inference

## Abstract

Large Language Models (LLMs) demonstrate exceptional capabilities across various tasks, but their deployment is constrained by high computational and memory costs. Model pruning provides an effective means to alleviate these demands. However, existing methods often ignore the characteristics of prefill-decode (PD) disaggregation in practice. In this paper, we propose a novel pruning method for PD disaggregation inference, enabling more precise and efficient block and KV Cache pruning. Our approach constructs pruning and distillation sets to perform iterative block removal independently for the prefill and decode stages, obtaining better pruning solutions. Moreover, we introduce a cache pruning mechanism that selectively reuses entries corresponding to the first and last token sequences within designated layers, reducing communication costs while incurring only negligible computational overhead. Extensive experiments demonstrate that our approach consistently achieves strong performance in both PD disaggregation and PD unified settings without disaggregation. Under the same (default) settings, our method achieves improved performance and faster inference, along with a $4.95\times$ reduction in data transmission bandwidth consumption.

## 1 Introduction

Large Language Models (LLMs) have emerged as revolutionary tools, achieving state-of-the-art (SOTA) results across diverse tasks and applications (Ding et al., 2022; Qin et al., 2023; Zhu et al., 2023; Li et al., 2023a). However, the rapid growth in model scale has posed significant challenges for their practical deployment (Zhang et al., 2023a; Choi et al., 2025; Long et al., 2025). To address it, various techniques have been proposed, including pruning (Ma et al., 2023; Ashkboos et al., 2024; Li et al., 2023c; Sun et al., 2025), quantization (Liu et al., 2021; Zhou et al., 2023b; Cai et al., 2023; Zhou et al., 2024) and knowledge distillation (Hinton et al., 2015; Gou et al., 2021; Yang et al., 2021; Zhang et al., 2024). Among them, pruning stands out as an effective strategy for reducing model size by eliminating redundant or less critical components, thereby lowering computational and storage costs.

While pruning offers clear benefits, its application to LLMs remains challenging. During inference, the prefill and decode stages are usually disaggregated (PD disaggregation) (Zhong et al., 2024; Patel et al., 2024; Qin et al., 2024; Dong et al., 2025), allowing each stage to be optimized according to its specific resource requirements. Without such disaggregation, heterogeneous workloads must conform to a single resource profile, which results in suboptimal utilization and ultimately degrades performance. However, existing pruning methods often ignore the characteristics of PD disaggregation deployment in real deployments. In such systems, pruning is confronted with two key challenges: **(1) Heterogeneous Pruning Sensitivity:** The prefill and decode stages exhibit markedly different sensitivities to pruning, making uniform strategies ineffective. **(2) Significant Bandwidth Overhead:** The physical disaggregation of prefill and decode nodes demands extensive data transfer, such as KV Cache, imposing significant communication costs.

Given these challenges, we propose addressing them through two complementary approaches: block and KV Cache pruning (each targeting a key point, rather than a simple combination). However, existing techniques are difficult to apply directly to PD disaggregation. (1) On one hand, current block pruning methods (Men et al., 2024; Kim et al., 2022; Yang et al., 2024b; Kim et al., 2024; Song et al., 2024) are unstable, and greedy selection strategies are limited to achieving local optimal

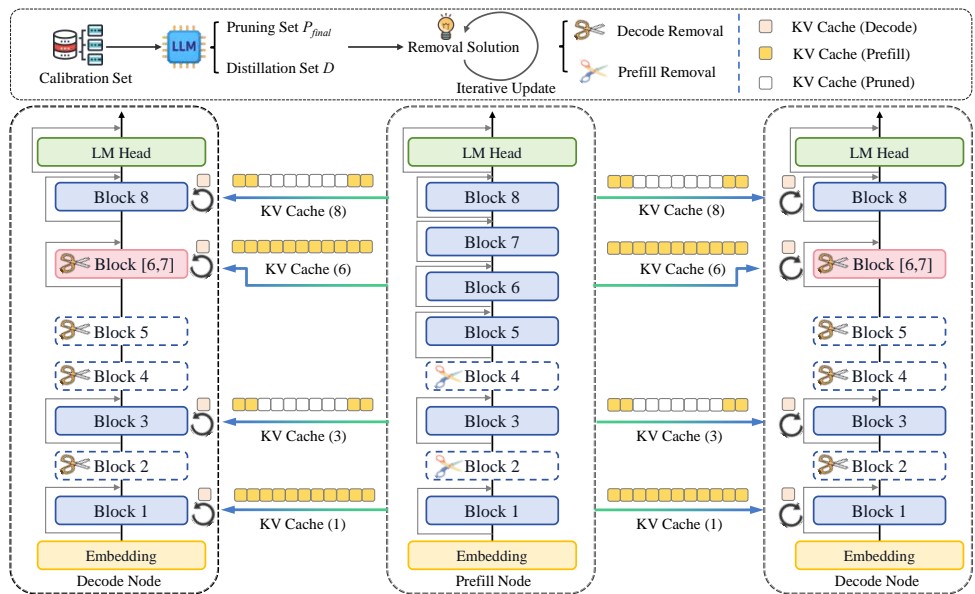

Figure 1: Overview of our pruning method combined with PD disaggregation. After execution on the prefill node, prompts are distributed to multiple decode nodes. Prefill and decode nodes use distinct block removal solutions, drawn from pruning and distillation sets. We select specific layers (such as 3 and 8), for which only partial KV Cache from the prefill node is transmitted, while the decode node KV Cache is fully preserved.

solutions. (2) On the other hand, existing KV Cache pruning methods (Xiao et al., 2023; Zhang et al., 2023b; Sun et al., 2024; Liu et al., 2024; Lee et al., 2024; Li et al., 2024) typically require retraining, add management overhead, and overlook the detailed perspective of attention heads, which limits throughput in PD disaggregation systems. Therefore, these fail to meet the deployment requirements of PD disaggregation systems.

In this paper, we propose a pruning method that is highly integrated with PD disaggregation, enabling more precise and efficient pruning of blocks and KV Cache. (1) Specifically, we propose an iterative block removal strategy for LLMs, guided by dedicated pruning and distillation sets. Our approach enables the identification of stage-specific removable blocks, independently tailored to the prefill and decode stages. Compared to prior block pruning approaches, our method achieves better solutions. (2) Moreover, we identify layers in the model at the granularity of attention heads where the sum of attention scores for the first and last token sequences is high. For the selected layers, we leverage the KV Cache only for the first and last token sequences. This efficiently reduces bandwidth consumption. Compared to prior approaches, our KV Cache pruning incurs negligible overhead. (3) Extensive experiments demonstrate that our approach consistently achieves strong performance in both PD disaggregation and PD unified (non-PD disaggregation) settings. Under the same (default) settings, our method achieves improved performance and faster inference, along with a $4.95\times$ reduction in data transmission bandwidth consumption.

Our contributions are as follows:

1. We introduce a novel pruning approach that is seamlessly integrated with PD disaggregation. This method facilitates more precise and efficient pruning of blocks and KV Cache. Extensive experiments demonstrate that our approach consistently achieves strong performance in both PD disaggregation and PD unified settings without disaggregation.

2. We construct pruning and distillation sets to perform iterative block removal, independently tailored to the prefill and decode stages, achieving superior performance over prior methods under identical settings.

3. We select layers with high combined scores corresponding to the first and last token sequences. For these layers, we selectively leverage partial KV Cache. This strategy introduces only negligible overhead compared to prior approaches.

## 2 METHOD

In this section, we describe our pruning method that is deeply integrated with PD disaggregation, allowing for more precise and efficient pruning of blocks and KV Cache. Figure 1 illustrates an overview of our method.

### 2.1 IMPLEMENTATION OF STAGE-AWARE STRATEGY FOR OPTIMAL BLOCK REMOVAL

In the following, we introduce an iterative block removal approach applied independently to the prefill and decode stages. Our goal is to select the $k$ blocks whose removal minimally impacts performance.

#### 2.1.1 STRATEGIC DESIGN OF SETS FOR PRUNING AND DISTILLATION

We use a combined pruning and distillation strategy for refined block removal. Specifically, we first assess the redundancy of each block by calculating the cosine similarity (Men et al., 2024) between its input and output (note that the similarity metric is not a core contribution of ours). We provide further details on this in Appendix I. The redundancy of $block_i$ is mathematically defined as follows:

$$r_i = \cos(h_{i-1}, h_i) \tag{1}$$

Here, $r_i$ denotes the redundancy of $block_i$ and the $\cos$ measures the similarity between the input $h_{i-1}$ and the output $h_i$ of $block_i$. Based on this redundancy metric, we directly identify the $Top \lceil \frac{k}{2} \rceil$ blocks with the highest cosine similarity for removal. Given their high redundancy, these blocks are deemed suitable for direct pruning, thereby forming an initial pruning set of $\lceil \frac{k}{2} \rceil$ blocks. The initial pruning set $\mathcal{P}_{initial}$ can be represented as follows:

$$\mathcal{P}_{initial} = \{block_i \mid r_i \in Top \lceil \tfrac{k}{2} \rceil (\{r_1, r_2, \ldots, r_L\})\} \tag{2}$$

For the blocks that are not included in the initial $\lceil \frac{k}{2} \rceil$ blocks designated for pruning, we consider incorporating them into the construction of the distillation set. To ensure the effectiveness of the distillation, we limit distillation to pairs of consecutive blocks. Specifically, for a pair of consecutive blocks, denoted as $block_i$ and $block_{i+1}$, we define a metric $d_i$ to determine whether they should be included in the distillation set. The metric $d_i$ is defined as follows:

$$d_i = \tfrac{1}{2} \left( \cos(h_{i-1}, h_{i+1}) + \max(\cos(h_{i-1}, h_i), \cos(h_i, h_{i+1})) \right) \tag{3}$$

This metric is designed from two complementary perspectives. Here, the $\cos(h_{i-1}, h_{i+1})$ measures redundancy when treating two blocks as a single unit; higher values imply weaker transformation and easier merging. During distillation, we merge consecutive blocks by finetuning the block with lower redundancy and transferring representational capacity from the block with higher redundancy. The more redundant block is identified by $\max(\cos(h_{i-1}, h_i), \cos(h_i, h_{i+1}))$; a larger value indicates higher redundancy and easier capacity transfer. We incorporate a pair of consecutive blocks into the distillation set and distill them into a single block only when the metric $d_i$ exceeds a predefined threshold $d_T$. The distillation set $\mathcal{D}$ can be represented as follows:

$$\mathcal{D} = \{(block_i, block_{i+1}) \mid d_i \geq d_T \ and \ \{block_i, block_{i+1}\} \cap \mathcal{P}_{initial} = \emptyset\} \tag{4}$$

If a block meets the threshold with both its preceding and succeeding blocks, we select the one with the higher metric to include in the distillation set. Through this approach, we construct the distillation set, where each element represents the distillation of a pair of consecutive blocks into one block. Once the distillation set is constructed, all blocks not included in it are assigned to the pruning set. The final pruning set $\mathcal{P}_{final}$ can be represented as follows:

$$\mathcal{P}_{final} = \{block_i \mid \{(block_{i-1}, block_i), (block_i, block_{i+1})\} \cap \mathcal{D} = \emptyset\} \tag{5}$$

In this setup, each block in the model is assigned to either the pruning set or the distillation set. Ingeniously, the number of blocks removed by each element in the distillation set matches that in the pruning set, with both strategies removing exactly one block. This consistency establishes a foundation for iterative optimization strategies.

### 2.1.2 ITERATIVE OPTIMIZATION TOWARD OPTIMAL BLOCK REMOVAL

To determine the optimal combination of block removals, we iteratively refine the removal set to minimize the impact on model performance. Specifically, we first initialize the block removal set by selecting the $Top \left\lceil \frac{k}{2} \right\rceil$ blocks with the highest cosine similarity from the pruning set. The remaining $k - \left\lceil \frac{k}{2} \right\rceil$ blocks $\mathcal{B}_{partial}$ are chosen from the distillation set based on the aforementioned metrics $d_i$ in descending order. If the number of blocks in the distillation set is insufficient, we supplement them from the pruning set in descending order of cosine similarity, excluding the $Top \left\lceil \frac{k}{2} \right\rceil$ blocks that need to be pruned. The initial block removal set $\mathcal{B}_{initial}$ can be represented as follows:

$$\mathcal{B}_{partial} = \begin{cases} Top(k - \lceil \frac{k}{2} \rceil)(\mathcal{D}) & if \ |\mathcal{D}| \geq k - \lceil \frac{k}{2} \rceil \\ \mathcal{D} \cup Top\left(k - \lceil \frac{k}{2} \rceil - |\mathcal{D}|\right)(\mathcal{P}_{final} \setminus \mathcal{P}_{initial}) & if \ |\mathcal{D}| < k - \lceil \frac{k}{2} \rceil \end{cases} \tag{6}$$

$$\mathcal{B}_{initial} = \mathcal{P}_{initial} \ \cup \ \mathcal{B}_{partial} \tag{7}$$

We define several key parameters for the optimization process. The initial temperature $T_0$ controls the randomness in the block iteration, the decay coefficient $\alpha$ is employed to gradually reduce the temperature and the termination temperature $T_{min}$ determines when to stop the entire block iteration (we provide the analysis of these parameters in experiments). In each iteration, we select an element from the unremoved set based on probability to replace a random element in the current removed set, thereby generating a new neighborhood solution. To determine the probability of each element being selected, we assign a weight $\omega_i$ to each element $i$ in the unremoved set. For every element $i$, $block_i$ in the pruning set or element $block_i$ and $block_{i+1}$ in the distillation set, the weight $\omega_i$ and the probability $p_i$ are related to formula 1, being selected can be defined as follows:

$$\omega_i = \begin{cases} r_i & pruning \ set \\ \frac{1}{2}(r_i + r_{i+1}) & distillation \ set \end{cases} \quad p_i = \frac{\omega_i}{\sum_{j=1}^{N} \omega_j} \tag{8}$$

Here, $N$ denotes the total number of elements in the unremoved set. For the new candidate block removal solution $s'$, we calculate its accuracy on the calibration set as the objective function value $f_{s'}$, and compare it with the current solution's ($s$) objective function value $f_s$. If the accuracy of the candidate solution is higher ( $f_{s'} > f_s$ ), we directly accept it. However, if the candidate solution's accuracy is lower ( $f_{s'} < f_s$ ), we decide whether to accept it based on a probability. The acceptance probability $P_{s'}$ is calculated using the following formula:

$$P_{s'} = e^{-\frac{\Delta f}{T}} \quad \Delta f = f_s - f_{s'} \tag{9}$$

where $\Delta f$ is the difference in objective function values between the current solution and the new candidate solution, and $T$ is the current temperature. This probability based acceptance mechanism allows us to escape local optima and explore the solution space more comprehensively. After evaluating the new solution, we update the temperature using the decay factor $T = \alpha T$. We repeat this process until the temperature reaches the minimum threshold $T_{min}$. Our experiments show that even with fewer iterations, performance still remains strong. Throughout the entire iterative process, we continuously record the block removal combination with the highest performance, considering it as the optimal solution. The optimal solution $\mathcal{B}_{optimal}$ can be represented as follows:

$$\mathcal{B}_{optimal} = \arg \max_{\mathcal{B} \in solutions} f_{\mathcal{B}} \tag{10}$$

where $f_{\mathcal{B}}$ is the objective function value of solution $\mathcal{B}$. This strategy yields better solutions that approximate the global optimum, as confirmed by experiments. It requires only a few iterations to achieve improved results, and the partial randomness introduced does not noticeably degrade performance. Moreover, it incurs substantially lower pruning overhead compared to other global search methods, as verified empirically.

### 2.1.3 HETEROGENEOUS STRATEGIES ACROSS PREFILL AND DECODE NODES

The prefill stage is more sensitive to pruning than the decode, pruning during this stage tends to lead to a more significant drop in performance. Here, we provide an abstract analysis, ignoring other complex effects. Denote by $X \in \mathbb{R}^{N \times d}$ the input sequence in the prefill stage and by $x_t \in \mathbb{R}^{1 \times d}$ the initial input in the decode stage. We denote the perturbations of the attention parameters $W_Q$,

$W_K$ and $W_V$ by $\Delta W_Q$, $\Delta W_K$ and $\Delta W_V$, respectively. The query vector $q$ and its perturbation $\Delta q$ for $x_t$ are as follows:

$$q = x_t W_Q, \ \Delta q = x_t \Delta W_Q \tag{11}$$

The concatenated key and value matrices $(K, V)$, along with their perturbations $(\Delta K, \Delta V)$, can be represented as follows:

$$K = [K_{pre}, \ x_t W_K], \ V = [V_{pre}, \ x_t W_V] \tag{12}$$

$$\Delta K = [\Delta K_{pre}, \ x_t \Delta W_K], \ \Delta V = [\Delta V_{pre}, \ x_t \Delta W_V] \tag{13}$$

Here, $K_{pre} = X W_K$ and $\Delta K = X \Delta W_K$; the same applies to the others. Then, the attention output before $(o)$ and after perturbation $(\tilde{o})$, as well as the resulting error $(E)$, can be expressed as follows:

$$o = softmax\left(\frac{qK^\top}{\sqrt{d}}\right) V, \ \tilde{o} = softmax\left(\frac{(q+\Delta q)(K+\Delta K)^\top}{\sqrt{d}}\right)(V + \Delta V) \tag{14}$$

$$\tilde{A} = softmax\left(\frac{(q+\Delta q)(K+\Delta K)^\top}{\sqrt{d}}\right), \ \Delta A = \tilde{A} - softmax\left(\frac{qK^\top}{\sqrt{d}}\right) \tag{15}$$

$$E = \tilde{o} - o = \Delta A \, V + \tilde{A} \, \Delta V \tag{16}$$

Considering the above analysis, we obtain the following formulas *(The detailed derivation of these formulas can be found in Appendix A and B):*

$$\|E\|_F \leq \frac{L_{softmax}}{\sqrt{d}}\left(\|\Delta q\|_2 \|K\|_F + \|q\|_2 \|\Delta K\|_F\right)\|V\|_F + \|\Delta V\|_F \tag{17}$$

$$E = G(x_t, Y_{pre}, \Delta) \tag{18}$$

Here, $L_{softmax}$ denotes the Lipschitz constant of the softmax, $\|E\|_F$ denotes the Frobenius norm of the error $E$, and $G$ is an abstract error function. $Y_{pre}$ denotes the set of all results (including both intermediate and final outputs) produced in the previous step that can be reused in the current step, and $\Delta$ represents the set of perturbations. From this, we observe that pruning errors accumulate as the sequence is generated. During the prefill stage, a perturbation simultaneously contaminates the representations of all $N$ tokens. These corrupted representations are repeatedly accessed in subsequent decode steps, causing the error to accumulate and amplify over time. In contrast, a single decode step only affects the current input and its subsequent states, limiting the scope of impact. Therefore, after determining the optimal combination of removed blocks, we further adjust the removed blocks for each stage. Specifically, for each element in the current combination of removed blocks, we test it on the calibration set. If removing only in the decode stage can significantly improve performance, then that element is removed only in the decode stage while being retained in the prefill stage. If there is no significant performance improvement, then the removing operation for that element will be applied in both the prefill and decode stages. More details can be found in Appendix M and C. This approach allows us to determine the removed blocks for the prefill and the decode separately.

## 2.2 Selective KV Cache Pruning for Reduced Bandwidth Consumption

In attention mechanisms, the highest scoring tokens are typically concentrated on the initial tokens and within a local sliding window (Xiao et al., 2023). However, prior approaches often introduce additional management overhead, making them unable to meet the high throughput demands of PD disaggregation systems. Based on this observation, we implement an efficient KV Cache pruning strategy that effectively reduces the transmission bandwidth consumption. This incurs negligible overhead and is robust across datasets, as confirmed by our experiments. We implement different strategies for the prefill and decode stages. During the prefill stage, the KV Cache is fully generated and utilized, whereas in the decode stage, only the first and last token sequences from the prefill stage are preserved for the selected layers. Specifically, we run inference on a calibration set to collect token attention scores for each attention head in every layer. We then calculate the sum of the attention scores for the first $p$ and last $p$ tokens of each attention head as the attention score metric (where $p$ denotes the proportion of tokens preserved from the beginning and the end). Mathematically, for each attention head $h$ in layer $l$, the attention score $S_h^{(l)}$ is given by:

$$S_h^{(l)} = \sum_{i=1}^{\lfloor p \cdot N \rfloor} A_h^{(l)}(i) + \sum_{j=N-\lfloor p \cdot N \rfloor + 1}^{N} A_h^{(l)}(j), \ p < 0.5 \tag{19}$$

where $A_h^{(l)}(i)$ denotes the attention score of the $i$-th token and $N$ is the total number of tokens. Our goal is to prune the KV Cache at the granularity of attention heads by selecting the top $n$ layers with the highest attention scores for the first and last token sequences. Meanwhile, we ensure that none of the attention heads in these layers have very low attention scores for the first and last token sequences. For the selected $n$ layers, we retain the KV Cache only for the first and last token sequences, while pruning the KV Cache for the remaining tokens. We first filter out any layers that contain attention heads with attention scores for the first and last token sequences below $\gamma$ (where $\gamma$ is the filtering threshold). Specifically, for each layer $l$, we check if all attention heads satisfy the condition:

$$\forall h \in \mathcal{H}^{(l)}, \quad S_h^{(l)} \geq \gamma \tag{20}$$

where $\mathcal{H}^{(l)}$ denotes the set of all attention heads in layer $l$. If a layer fails to meet this criterion, it is subsequently excluded from further consideration. Then, we will use the following scoring formula to calculate the score $\rho_l$ for each layer $l$:

$$\rho_l = \mu_l \cdot (1 - \sigma_l/(\mu_l + \epsilon)) \tag{21}$$

Here, $\mu_l$ and $\sigma_l$ denote the mean and standard deviation of all attention heads in layer $l$, and $\epsilon$ is an infinitesimal constant. This formula requires both high mean values and low standard deviations to be satisfied. We select the top layers with the highest scores in descending order. In practical deployment, we generate and use the full KV Cache during the prefill stage. In decode stage, only the KV Cache of the first and last token sequences from prefill is reused for selected layers, significantly reduces the bandwidth usage for data transfer from the prefill nodes to the decode nodes.

The primary goal of our KV Cache pruning is to reduce communication bandwidth consumption. As such, pruning is applied only to the KV Cache generated on the prefill node, while the cache generated on the decode node is fully preserved. We divide the generation process into three stages for the selected layers. Initial Stage: The tokens with the highest attention scores correspond closely to the KV Cache retained on the prefill node. Intermediate Stage: As the generated sequence lengthens, the tokens with the highest attention scores become the first and last several tokens retained on the prefill node, along with the newly generated tokens on the decode node. Late Stage: As the generated sequence further grows, attention scores concentrate on the first several tokens retained on the prefill node, and the latest several tokens generated on the decode node.

## 3 EXPERIMENTS

### 3.1 EXPERIMENTAL SETUP

**Models and Benchmarks** To demonstrate the effectiveness of our method, we conduct extensive evaluations on representative LLMs with diverse architectures and scales, including LLaMA3.1-8B (Grattafiori et al., 2024), LLaMA2-13B (Touvron et al., 2023), Qwen2.5-7B (Yang et al., 2024a) and Qwen2.5-14B (Yang et al., 2024a). We also use Qwen2.5-32B (Yang et al., 2024a) and OPT-6.7B (Zhang et al., 2022) in the supplementary experiments. We employ a wide range of benchmarks. These benchmarks include MMLU (Hendrycks et al., 2020), CMMLU (Li et al., 2023b), PIQA (Bisk et al., 2020), Winogrande (ai2, 2019), HellaSwag (Zellers et al., 2019), BoolQ (Clark et al., 2019), MathQA (Amini et al., 2019), ARC-Easy and ARC-Challenge (Clark et al., 2018), RTE (Wang, 2018), WNLI (Wang, 2018), CB (Wang et al., 2019) and SST-2 (Wang, 2018). This comprehensive protocol ensures thorough assessment.

**Baselines** We conduct comparative evaluations against other methods, including LLM-Pruner (Ma et al., 2023), FLAP (An et al., 2024), Shortened LLaMA (abbreviated as Shortened)(Kim et al., 2024), ShortGPT (Men et al., 2024) and SLEB (Song et al., 2024). Additionally, we also use SliceGPT (Ashkboos et al., 2024) to test inference speed. We also compare with other methods in Appendix G. We implement PD disaggregation versions of the classic channel pruning LLM-Pruner and block pruning ShortGPT according to our algorithm, and use them as additional baselines. These comparisons highlight the strengths of our approach. More details can be found in Appendix C.

**Implementation Details** Our experiments are conducted using the PyTorch framework (Paszke et al., 2019) and the Hugging Face Transformers library (Wolf, 2020). We use two nodes within the same local area network, with each node equipped with one NVIDIA H100 80GB GPU. When distilling two blocks, we use the weights of the block with low cosine similarity as the initial weights.

Table 1: Performance comparison of pruning methods in LLMs across a variety of benchmarks. To increase the diversity of evaluation, we apply pruning rates of 13.6%, 24.4%, 15.3% and 18.6% for the four models, corresponding to commonly used pruning ratios for structured pruning.

| LLM | Method | MMLU | CMMLU | ARC-E | ARC-C | PIQA | Winog | HSwag | BoolQ | MathQA | WNLI | SST-2 | RTE | CB | AVG |
|---|---|---|---|---|---|---|---|---|---|---|---|---|---|---|---|
| LLaMA3.1-8B | Dense | 63.35 | 50.85 | 81.52 | 51.28 | 80.30 | 74.03 | 60.05 | 82.26 | 39.56 | 59.15 | 76.83 | 71.12 | 60.71 | 65.46 |
| | LLM-Pruner | 52.01 | 41.12 | 67.36 | 42.06 | 74.54 | 69.32 | 51.50 | 73.78 | 32.35 | 50.78 | 69.37 | 49.40 | 55.28 | 56.07 |
| | FLAP | 52.11 | 41.13 | 67.42 | 42.15 | 74.63 | 69.36 | 51.56 | **73.82** | 32.39 | 50.82 | 69.41 | 49.44 | 55.34 | 56.11 |
| | Shortened | 33.54 | 34.28 | 72.31 | 42.15 | 72.63 | 69.61 | 47.96 | 45.23 | 34.27 | 56.34 | 52.52 | 67.15 | 69.64 | 53.66 |
| | ShortGPT | 51.92 | 41.11 | 67.34 | 42.06 | 74.48 | 69.38 | 51.56 | 73.76 | 32.50 | 50.70 | 69.31 | 49.50 | 55.36 | 56.08 |
| | SLEB | 28.35 | 25.51 | 71.04 | 36.18 | 75.46 | 62.98 | 49.70 | 57.89 | 27.30 | 46.48 | 55.85 | 57.04 | 35.71 | 48.42 |
| | Ours | **60.82** | **46.66** | **73.15** | **43.86** | **76.99** | **73.56** | **53.80** | 69.85 | **34.28** | **60.56** | **83.03** | **70.76** | **73.21** | **62.99** |
| LLaMA2-13B | Dense | 52.10 | 34.73 | 79.42 | 48.38 | 79.05 | 72.22 | 60.07 | 80.61 | 32.09 | 66.20 | 87.61 | 69.31 | 80.36 | 64.78 |
| | LLM-Pruner | 50.11 | 33.57 | 61.16 | 37.67 | 71.38 | 70.71 | 47.37 | 62.53 | 24.41 | 43.24 | 65.37 | 59.19 | **51.38** | 52.16 |
| | FLAP | 49.89 | 33.91 | 60.90 | 37.65 | 71.44 | 70.69 | 47.54 | 62.43 | 24.78 | 43.26 | 64.91 | 59.19 | 51.34 | 52.15 |
| | Shortened | 26.71 | 25.50 | 26.26 | 22.61 | 51.14 | 48.22 | 25.76 | 38.93 | 18.89 | 43.66 | 46.56 | 52.71 | 37.50 | 35.73 |
| | ShortGPT | 50.13 | 33.97 | 61.32 | 37.88 | 71.44 | **70.80** | 47.71 | 62.54 | 24.82 | 43.66 | 65.37 | 59.57 | 51.79 | 52.38 |
| | SLEB | 23.76 | 25.41 | 67.85 | 33.87 | **75.41** | 63.77 | 48.76 | 62.42 | 25.46 | 45.07 | 50.92 | 58.84 | 41.07 | 47.89 |
| | Ours | **51.52** | **35.37** | **69.78** | **40.02** | 74.43 | 70.72 | **53.11** | **62.84** | **28.44** | 53.52 | **80.16** | **64.26** | 46.43 | **56.20** |
| Qwen2.5-7B | Dense | 71.92 | 81.69 | 80.56 | 48.21 | 78.73 | 72.93 | 59.92 | 84.46 | 43.22 | 71.83 | 91.86 | 81.23 | 87.50 | 73.39 |
| | LLM-Pruner | 36.96 | 36.71 | 71.33 | 38.20 | 74.86 | 55.79 | 47.71 | 55.67 | 31.08 | 60.43 | 70.79 | 54.42 | 38.81 | 51.75 |
| | FLAP | 37.17 | 36.39 | 71.47 | 37.83 | 76.21 | 55.65 | 47.47 | 57.23 | 30.11 | 61.61 | 70.68 | 54.59 | 37.32 | 51.82 |
| | Shortened | 24.90 | 25.08 | 25.25 | 20.31 | 53.65 | 50.99 | 25.69 | 37.83 | 19.50 | 53.52 | 50.92 | 46.21 | 19.64 | 34.88 |
| | ShortGPT | 36.89 | 31.07 | 71.46 | 37.80 | 76.12 | 55.80 | 48.67 | 63.21 | 30.59 | 42.25 | **80.62** | 54.87 | 41.07 | 51.57 |
| | SLEB | 38.56 | 38.25 | **71.84** | **39.42** | 76.77 | 55.96 | 47.98 | 57.49 | **31.12** | 61.97 | 72.25 | 56.32 | 39.29 | 52.86 |
| | Ours | **52.96** | **51.34** | 71.09 | 39.16 | **77.04** | **56.99** | **51.19** | **72.20** | 30.95 | **66.20** | 63.53 | **65.34** | **57.14** | **58.09** |
| Qwen2.5-14B | Dense | 77.45 | 84.44 | 82.37 | 56.31 | 81.12 | 75.37 | 63.37 | 85.23 | 53.03 | 77.46 | 89.11 | 79.78 | 80.36 | 75.80 |
| | LLM-Pruner | 43.09 | 42.01 | 73.33 | 40.54 | 73.58 | 58.47 | 47.79 | 61.89 | 31.53 | 48.79 | 54.26 | 56.88 | 49.27 | 52.42 |
| | FLAP | 44.86 | 44.77 | 50.29 | 31.00 | 60.72 | 51.44 | 34.14 | 65.16 | 25.65 | 65.28 | 89.95 | 76.87 | 68.74 | 54.53 |
| | Shortened | 24.63 | 25.31 | 25.04 | 20.14 | 52.88 | 50.43 | 25.69 | 37.92 | 18.93 | 52.11 | 48.62 | 51.99 | 37.50 | 36.25 |
| | ShortGPT | 45.75 | 45.63 | 50.63 | 31.40 | 61.81 | 52.64 | 34.41 | **65.72** | 26.16 | 66.20 | **91.28** | **78.34** | 69.64 | 55.35 |
| | SLEB | 43.77 | 42.90 | **74.07** | 41.04 | 74.37 | 58.96 | **48.43** | 62.57 | 32.23 | 49.30 | 54.93 | 57.40 | 50.00 | 53.07 |
| | Ours | **72.01** | **76.82** | 68.81 | **44.37** | 71.38 | **70.17** | 47.98 | 64.80 | **34.91** | 61.97 | 82.11 | 74.01 | **69.65** | **64.54** |

By default, we use 256 randomly sampled examples from PIQA and MMLU as the calibration set. The hyperparameter settings and training details can be found in Appendix C.

## 3.2 MAIN RESULTS

We evaluate our method against strong baselines on four representative LLMs and multiple benchmarks under identical experimental settings, including the calibration set. To diversify the evaluation, we adopt pruning ratios of 13.6%, 24.4%, 15.3% and 18.6% across the four models. The Dense configuration corresponds to the original uncompressed model, serving as a reference prior to pruning. The experiments are conducted with consistent parameter settings and implementation details. As shown in Table 1, our method consistently outperforms all baselines across different models. These results clearly confirm the effectiveness and robustness of our method under diverse pruning scenarios, ensuring broad applicability and reliable performance.

Our approach serves as an optimization for PD disaggregation and can be readily combined with other pruning methods. We extend two representative approaches, the channel pruning method LLM-Pruner and the block pruning method ShortGPT. Specifically, we incorporate our proposed PD disaggregation scheme into these methods, denoted as LLM-Pruner (Ours) and ShortGPT (Ours). As shown in Table 2, both methods exhibit substantial performance improvements on the evaluated datasets after applying PD disaggregation. Moreover, our approach consistently achieves the best overall results. These findings highlight the effectiveness of PD disaggregation as a general principle for enhancing pruning strategies in LLMs.

To validate the scalability and generality of our method, we conduct experiments on larger model, additional datasets, and an extra metric (perplexity), achieving promising results (see Appendix D.1, G). We further compare our approach against various baselines in the PD unified setting, where it still remains competitive (see Appendix D.3). Additionally, we evaluate our method under both PD disaggregation and unified settings, confirming the effectiveness of PD disaggregation in pruning (see Appendix D.4). Comparisons with other KV Cache pruning methods further show that our approach consistently outperforms alternatives (see Appendix G). Collectively, these results demonstrate that our method achieves efficient pruning while preserving representation capability.

## 3.3 EFFICIENCY ANALYSIS

Table 2: Performance comparison of our method with LLM-Pruner and ShortGPT, as well as their variants. LLM-Pruner (Ours) and ShortGPT (Ours) are our PD disaggregation extensions.

| LLM | Method | MMLU | CMMLU | ARC-E | ARC-C | HSwag | MathQA | WNLI | RTE | AVG |
|---|---|---|---|---|---|---|---|---|---|---|
| **LLaMA3.1-8B** | LLM-Pruner | 52.01 | 41.12 | 67.36 | 42.06 | 51.50 | 32.35 | 50.78 | 49.40 | 48.32 |
| | ShortGPT | 51.92 | 41.11 | 67.34 | 42.06 | 51.56 | 32.50 | 50.70 | 49.50 | 48.34 |
| | LLM-Pruner (Ours) | 54.95 | 44.30 | 72.46 | 43.82 | 52.84 | 34.13 | 53.94 | 52.88 | 51.16 |
| | ShortGPT (Ours) | 54.62 | 43.89 | 71.38 | 43.11 | 53.55 | **35.08** | 53.62 | 52.63 | 50.98 |
| | Ours | **60.82** | **46.66** | **73.15** | **43.86** | **53.80** | 34.28 | **60.56** | **70.76** | **55.49** |
| **LLaMA2-13B** | LLM-Pruner | 50.11 | 33.57 | 61.16 | 37.67 | 47.37 | 24.41 | 43.24 | 59.19 | 44.59 |
| | ShortGPT | 50.13 | 33.97 | 61.32 | 37.88 | 47.71 | 24.82 | 43.66 | 59.57 | 44.88 |
| | LLM-Pruner (Ours) | 52.58 | 34.27 | 65.55 | 39.19 | 49.99 | 25.96 | 44.53 | 62.54 | 46.83 |
| | ShortGPT (Ours) | **53.70** | 34.28 | 64.25 | 38.98 | 50.76 | 26.50 | 46.84 | 63.41 | 47.34 |
| | Ours | 51.52 | **35.37** | **69.78** | **40.02** | **53.11** | **28.44** | **53.52** | **64.26** | **49.50** |
| **Qwen2.5-7B** | LLM-Pruner | 36.96 | 36.71 | 71.33 | 38.20 | 47.71 | 31.08 | 60.43 | 54.42 | 47.11 |
| | ShortGPT | 36.89 | 31.07 | 71.46 | 37.80 | 48.67 | 30.59 | 42.25 | 54.87 | 44.20 |
| | LLM-Pruner (Ours) | 38.99 | 39.42 | 72.41 | 38.66 | 50.54 | **33.34** | 61.89 | 56.83 | 49.01 |
| | ShortGPT (Ours) | 39.51 | 33.01 | **73.04** | 39.01 | 51.18 | 32.88 | 62.96 | 56.64 | 48.53 |
| | Ours | **52.96** | **51.34** | 71.09 | **39.16** | **51.19** | 30.95 | **66.20** | **65.34** | **53.53** |
| **Qwen2.5-14B** | LLM-Pruner | 43.09 | 42.01 | 73.33 | 40.54 | 47.79 | 31.53 | 48.79 | 56.88 | 48.00 |
| | ShortGPT | 45.75 | 45.63 | 50.63 | 31.40 | 34.41 | 26.16 | 66.20 | 78.34 | 47.32 |
| | LLM-Pruner (Ours) | 65.75 | 45.32 | 74.49 | 43.13 | 48.75 | 32.43 | 51.27 | 58.44 | 52.45 |
| | ShortGPT (Ours) | 68.34 | 47.36 | 52.21 | 33.16 | 36.37 | 28.05 | **67.64** | **78.89** | 51.50 |
| | Ours | **72.01** | **76.82** | 68.81 | **44.37** | 47.98 | **34.91** | 61.97 | 74.01 | **60.11** |

**Inference Latency** We measure the execution time for a single inference of the LLaMA3.1-8B model with an input tensor of shape $[1, 1024]$ in FP32 format. Specifically, we evaluate the effects of pruning 13.58% and approximately 50% of the parameters in LLaMA3.1-8B. We then compare our proposed method against several other widely used pruning algorithms, including SliceGPT, LLM-Pruner and FLAP. As shown in Table 3, our approach consistently achieves the fastest runtime, significantly reducing inference latency. This demonstrates that our block pruning approach enables faster inference speed.

Table 3: Comparison of inference time (ms) for the LLaMA3.1-8B on different pruning methods.

| Method | Dense | SliceGPT | LLM-Pruner | FLAP | Ours |
|---|---|---|---|---|---|
| Time (13.58%) | 287.35 | 263.35 | 278.19 | 259.81 | 236.63 |
| Time (~50%) | 287.35 | 202.69 | 252.69 | 184.16 | 151.89 |

**Bandwidth Consumption** To provide a comprehensive evaluation, we further assess LLaMA3.1-8B and LLaMA2-13B under default settings with batch size 1 and maximum input length. Our primary focus is on the communication cost between prefill and decode nodes, a critical yet often overlooked aspect of distributed inference. To evaluate this, we carefully measure both the data volume and transmission time, allowing for a precise and comprehensive assessment of bandwidth usage. As shown in Table 4, our method substantially reduces data transmission and overall bandwidth consumption, highlighting its effectiveness in mitigating communication overhead during inference.

Table 4: Comparison of data transmission volume and transmission time for LLaMA series models.

| Model | Method | DataVol (G) | Time ($\mu s$) |
|---|---|---|---|
| LLaMA3.1-8B | Original | 4.0 | 11628 |
| | Ours | 0.8 | 2347 |
| LLaMA2-13B | Original | 7.0 | 20277 |
| | Ours | 1.4 | 4072 |

**Pruning Overhead** We conduct a comparison of the computational overhead between our proposed pruning method and SLEB. SLEB (Song et al., 2024) is also a pruning strategy that aims to identify the optimal pruning solution; however, it requires performing multiple calibration set evaluations in each pruning iteration, which significantly increases its computational cost. We prune five blocks on LLaMA3.1-8B. The hyperparameters we use have been provided in the experiment details in the previous section. As shown in Table 5, our method achieves an average runtime of only 44.06 seconds, whereas SLEB requires 183.6 seconds on average. Moreover, we conduct a comparison of pruning overhead on LLaMA2-

Table 5: Comparison of pruning time between our method and SLEB on LLaMA series models.

| Model | Method | Time (s) |
|---|---|---|
| LLaMA3.1-8B | SLEB | 183.6 |
| | Ours | 44.06 |
| LLaMA2-13B | SLEB | 858.12 |
| | Ours | 96.52 |

13B under a pruning rate of 24.4%, where our approach continues to demonstrate a significant advantage. This efficiency improvement is attributed to our iterative pruning strategy, which can more rapidly converge to an effective pruning configuration.

We further evaluate the latency of the calibration process on the LLaMA series models, where our calibration requires only minimal computational time (see Appendix J.2).

## 4 RELATED WORK

**PD Disaggregation** LLM inference can be divided into the prefill and decode stages. The prefill stage processes the entire input sequence to compute contextual representations and generate KV Cache. The decode stage then autoregressively generates tokens conditioned on previously generated outputs. PD disaggregation is a technique that explicitly decouples these two stages during inference, enabling deployment strategies tailored to the unique resource requirements of each stage. DistServe (Zhong et al., 2024) is one of the seminal works in PD disaggregation. It assigns the prefill and decode stages to different GPUs, allowing for optimizations tailored to the unique characteristics of each stage. Splitwise (Patel et al., 2024) disaggregates the prompt computation and token generation stages and runs them on different machines. The TetriInfer (Hu et al., 2024b) also leverages PD disaggregation to reduce interference between different downstream tasks. MemServe (Hu et al., 2024a) further explores memory optimization techniques within the PD disaggregation architecture. TaiChi (Wang et al., 2025) integrates PD disaggregation and aggregation by leveraging differentiated GPU roles and adaptive scheduling to optimize goodput across diverse SLO regimes. Adrenaline (Liang et al., 2025) offloads part of decode phase attention computation to prefill instances, improving LLM serving. The widespread application of PD disaggregation drives ongoing improvements in LLM inference efficiency.

**Block Pruning** To develop a straightforward pruning algorithm that is easy to deploy for LLMs, several studies have proposed removing less important blocks, a strategy that can lead to significant inference acceleration and improved computational efficiency. For example, methods (Men et al., 2024) using cosine similarity to evaluate block importance combined with greedy pruning have been introduced. However, greedy approaches often fail to identify the globally optimal pruning configuration and tend to exhibit instability in practice. Techniques like LaCo (Yang et al., 2024b) merge subsequent layers into preceding ones but typically sacrifice accuracy compared to direct layer removal. LLM-Streamline (Chen et al., 2024) reduces model size by distilling multiple consecutive blocks into a single block. However, overly aggressive distillation can significantly degrade model performance, and the approach still relies on a greedy block selection strategy. Other methods, such as SLEB (Song et al., 2024) and Shortened LLaMA (Kim et al., 2024), adopt iterative pruning strategies guided by carefully designed importance metrics. However, these methods require recalculating importance scores using a calibration set after each block removal, resulting in substantial computational overhead. Consequently, developing a block pruning algorithm that is both highly accurate and computationally efficient remains a difficult and unresolved challenge.

## 5 CONCLUSION

In this paper, we propose a pruning method that is deeply integrated with PD disaggregation. Our design explicitly takes into account the practical challenges and constraints that arise when deploying LLMs. In particular, we construct pruning and distillation sets to perform iterative block removal, independently tailored to the prefill and decode, achieving better solutions compared to prior block pruning approaches. Moreover, we select layers with high combined scores for the first and last token sequences. The prefill stage generates and utilizes all KV Cache, while the decode stage accesses only part of the KV Cache in the selected layers, reducing bandwidth usage. This incurs negligible overhead compared to prior methods. Under the same (default) settings, our method achieves improved performance and faster inference. Extensive experiments demonstrate that our approach consistently achieves strong performance in both PD disaggregation and PD unified settings without disaggregation, achieving effective inference acceleration and reduced bandwidth consumption.

ETHICS STATEMENT

This work adheres to the ICLR Code of Ethics. Our study does not involve human subjects, nor personal or sensitive data. All datasets utilized in this paper are publicly available and widely adopted within the research community, and we strictly follow their respective licenses and intended usage.

REPRODUCIBILITY STATEMENT

We strive to ensure the reproducibility of our results. Full details are provided in the main paper and the appendix. Our implementation is built on PyTorch and standard open-source libraries. We provide key code implementations to facilitate reproducibility and further research.

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

# A DETAILED DERIVATION OF THE THEORETICAL FORMULA

Denote by $X \in \mathbb{R}^{N \times d}$ the input sequence in the prefill stage and by $x_t \in \mathbb{R}^{1 \times d}$ the initial input in the decode stage. We denote the perturbations of the attention parameters $W_Q$, $W_K$ and $W_V$ by $\Delta W_Q$, $\Delta W_K$ and $\Delta W_V$, respectively. The query vector $q$ and its perturbation $\Delta q$ for $x_t$ are as follows:

$$q = x_t W_Q, \ \Delta q = x_t \Delta W_Q \tag{22}$$

The concatenated key and value matrices $(K, V)$, along with their perturbations $(\Delta K, \Delta V)$, can be represented as follows:

$$K = [K_{pre}, x_t W_K], \ V = [V_{pre}, x_t W_V] \tag{23}$$

$$\Delta K = [\Delta K_{pre}, x_t \Delta W_K], \ \Delta V = [\Delta V_{pre}, x_t \Delta W_V] \tag{24}$$

Here, $K_{pre} = X W_K$ and $\Delta K = X \Delta W_K$; the same applies to the others. Then, the attention output before $(o)$ and after perturbation $(\tilde{o})$, as well as the resulting error $(E)$, can be expressed as follows:

$$o = softmax\left(\frac{qK^\top}{\sqrt{d}}\right)V, \ \tilde{o} = softmax\left(\frac{(q+\Delta q)(K+\Delta K)^\top}{\sqrt{d}}\right)(V + \Delta V) \tag{25}$$

$$\tilde{A} = softmax\left(\frac{(q+\Delta q)(K+\Delta K)^\top}{\sqrt{d}}\right), \ \Delta A = \tilde{A} - softmax\left(\frac{qK^\top}{\sqrt{d}}\right) \tag{26}$$

$$E = \tilde{o} - o = \Delta A \, V + \tilde{A} \, \Delta V \tag{27}$$

To bound the Frobenius norm of the attention output perturbation $E$, we apply the triangle inequality to separate the contributions from the two terms, yielding the following expression:

$$\|E\|_F \leq \|\Delta A \, V\|_F + \|\tilde{A} \, \Delta V\|_F \tag{28}$$

Then, using the submultiplicativity of matrix norms, we can directly obtain the following bounds:

$$\|\Delta A \, V\|_F \leq \|\Delta A\|_F \|V\|_F, \ \|\tilde{A} \, \Delta V\|_F \leq \|\tilde{A}\|_2 \|\Delta V\|_F \tag{29}$$

By the Lipschitz continuity of the softmax, the perturbation $\Delta A$ can naturally be bounded as follows:

$$\|\Delta A\|_F \leq L_{softmax} \frac{\|(q + \Delta q)(K + \Delta K)^\top - qK^\top\|_F}{\sqrt{d}} \tag{30}$$

Here, $L_{softmax}$ is the Lipschitz constant of the softmax. Upon expanding the product, we can explicitly obtain the resulting expression as follows:

$$(q + \Delta q)(K + \Delta K)^\top - qK^\top = \Delta q \, K^\top + q \, (\Delta K)^\top + (\Delta q)(\Delta K)^\top \tag{31}$$

where the last term $(\Delta q)(\Delta K)^\top$ is of second order and can be neglected in a first order analysis. Applying the submultiplicativity of the Frobenius norm, we can further obtain the following bounds:

$$\|\Delta q \, K^\top\|_F \leq \|\Delta q\|_2 \|K\|_F, \ \|q(\Delta K)^\top\|_F \leq \|q\|_2 \|\Delta K\|_F \tag{32}$$

Considering all above analysis, we can express the Frobenius norm of the error $E$ as follows:

$$\|E\|_F \leq \frac{L_{softmax}}{\sqrt{d}} \left(\|\Delta q\|_2 \|K\|_F + \|q\|_2 \|\Delta K\|_F\right) \|V\|_F + \|\tilde{A}\|_2 \|\Delta V\|_F \tag{33}$$

Considering that $\|\tilde{A}\|_2 \leq 1$, we obtain the following expression:

$$\|E\|_F \leq \frac{L_{softmax}}{\sqrt{d}} \left(\|\Delta q\|_2 \|K\|_F + \|q\|_2 \|\Delta K\|_F\right) \|V\|_F + \|\Delta V\|_F \tag{34}$$

Building on the above formula, and to more clearly illustrate the effect of pruning, we can express the error $E$ at step $t$ using an abstract error function $G$ as follows:

$$E = G(x_t, Y_{pre}, \Delta) \tag{35}$$

where $Y_{pre}$ denotes the set of all results (including both intermediate and final outputs) produced in the previous step that can be reused in the current step, and $\Delta$ represents the set of perturbations.

## B    Frobenius Norm of $\Delta A$ via Softmax Lipschitz Continuity

For any $a, b \in \mathbb{R}^n$, the softmax function is Lipschitz continuous with respect to the Frobenius norm, which can be expressed as follows:

$$\|softmax(a) - softmax(b)\|_F \leq L_{softmax}\|a - b\|_F \tag{36}$$

Here, $L_{softmax}$ is the Lipschitz constant of the softmax. Based on the derivations in the main text, we obtain the following expression:

$$\Delta A = softmax\left(\frac{(q + \Delta q)(K + \Delta K)^\top}{\sqrt{d}}\right) - softmax\left(\frac{qK^\top}{\sqrt{d}}\right) \tag{37}$$

By jointly considering Equations 36 and 37, we can derive the following expression:

$$\|\Delta A\|_F \leq L_{softmax}\frac{\|(q + \Delta q)(K + \Delta K)^\top - qK^\top\|_F}{\sqrt{d}} \tag{38}$$

## C    Experimental Setup

**Baselines** We conduct extensive comparative evaluations against other pruning algorithms. LLM-Pruner (Ma et al., 2023) adopts structural pruning that selectively removes non-critical coupled structures based on gradient information. FLAP (An et al., 2024) is a structured pruning framework that reduces storage by leveraging fluctuation based metrics and adaptive model compression. Shortened LLaMA (Kim et al., 2024) selectively removes less important blocks based on block level importance scores, thereby accelerating model inference without significantly impacting performance. ShortGPT (Men et al., 2024) defines a BI metric to measure the importance of each layer within the model and directly removes those layers. SLEB (Song et al., 2024) employs a logit based approach to identify unnecessary transformer layers and updates the importance scores after each layer removal. SliceGPT (Ashkboos et al., 2024) is a post-training sparsification scheme which replaces each weight matrix with a smaller matrix. Through these comprehensive comparisons, we thoroughly assess the strengths of our approach.

**Implementation Details** Our experiments are conducted using the PyTorch framework (Paszke et al., 2019) and the Hugging Face Transformers library (Wolf, 2020). We use two nodes within the same local area network, with each node equipped with one NVIDIA H100 80GB GPU. We follow the respective compression strategies for prefill and decode as mentioned above. We set the parameter $d_T$ to 0.95 by default. During the iterative block selection process, we initialize the annealing temperature $T_0$ at 15 to ensure sufficient exploration in the early stages. The temperature is then gradually reduced with a decay coefficient $\alpha = 0.85$ until it reaches the minimum temperature $T_{\min} = 0.05$. When distilling two blocks, we use the weights of the block with low cosine similarity as the initial weights. We distill the block using the PIQA and MMLU as the training set. The training is performed using the Adam optimizer with a learning rate of $1 \times 10^{-5}$, a batch size of 64 for 10 epochs. When sparsifying the KV Cache, we set the pruning ratio $p = 0.3$, with an attention score threshold $\gamma = 0.75$ to preserve the most semantically important key value pairs. We set the performance improvement threshold mentioned in Section 2.1.3 to 3%. We conduct experiments with various hyperparameters to demonstrate the stability of our method. By default, we use 256 randomly sampled examples from PIQA and MMLU as the calibration set. In the zero shot performance comparisons, we maintain consistent experimental settings, including the calibration datasets.

## D    Experiments on Strong Scalability and Robust Generality

### D.1    Larger Model, More Datasets and Additional Metric

We further evaluate our method on the larger Qwen2.5-32B model with a pruning ratio of 25%. As reported in Table 6, our approach consistently achieves strong performance. Additionally, we report perplexity on WikiText2 using LLaMA2-13B and OPT-6.7B with pruning ratios of 24.37% and 20%, respectively, where our method achieves the best results (Table 7). Beyond these evaluations, we assess LLaMA3.1-8B-Instruct (Grattafiori et al., 2024) under a pruning ratio of 15.3% on two

further benchmarks: the instruction-following dataset IFEval (Zhou et al., 2023a) and the code generation dataset HumanEval (Chen et al., 2021). As shown in Table 8, our method maintains strong performance across these tasks.

Table 6: Performance comparison of various methods on Qwen2.5-32B under 25% pruning.

| Method | PIQA | Winog | HSwag | ARC-E | ARC-C |
|---|---|---|---|---|---|
| Dense | 81.88 | 75.3 | 64.91 | 80.51 | 53.41 |
| LLM-Pruner | 78.02 | 61.64 | 55.02 | 70.24 | 42.24 |
| FLAP | 77.56 | 61.08 | 55.23 | 70.36 | 42.37 |
| Shortened | 76.55 | 61.81 | 55.39 | 71.31 | 41.7 |
| ShortGPT | 76.66 | 72.53 | 52.4 | 72.85 | 42.24 |
| SLEB | 72.28 | 65.88 | 56.32 | 69.22 | 38.61 |
| Ours | **78.45** | **72.85** | **57** | **75.8** | **46.67** |

Table 7: Evaluation of perplexity on WikiText2 for LLaMA2-13B and OPT-6.7B with pruning.

| Model | Dense | LLM-Pruner | FLAP | Shortened | ShortGPT | SLEB | Ours |
|---|---|---|---|---|---|---|---|
| OPT-6.7B | 10.86 | 11.92 | 11.68 | 12.51 | 13.68 | 12.94 | **11.28** |
| LLaMA2-13B | 4.88 | 8.16 | 9.73 | 10.81 | 9.25 | 8.96 | **7.38** |

Table 8: Evaluation of LLaMA3.1-8B-Instruct under 15.3% pruning on IFEval and HumanEval.

| Benchmark | Dense | LLM-Pruner | FLAP | Shortened | ShortGPT | SLEB | Ours |
|---|---|---|---|---|---|---|---|
| IFEval | 76.36 | 65.41 | 65.45 | 62.60 | 65.42 | 56.48 | **73.48** |
| HumanEval | 68.95 | 59.36 | 60.26 | 57.27 | 60.44 | 51.39 | **66.99** |

## D.2 ORTHOGONAL TO QUANTIZATION

To evaluate the compatibility and effectiveness of our method under combined compression strategies, we further integrate it with quantization techniques. Specifically, we apply our approach to the Qwen2.5-32B model with a pruning ratio of 25% in conjunction with 8-bit AWQ quantization (Lin et al., 2024). As summarized in Table 9, our method consistently demonstrates robust performance, indicating that the pruning strategy is largely orthogonal to quantization and can be effectively combined without significant degradation.

## D.3 PERFORMANCE COMPARISON IN THE NON-PD DISAGGREGATION SETTING

We compare the performance of our method with other approaches in the non-PD disaggregation setting, referred to as PD Unified. Specifically, we evaluate our method on the LLaMA2-13B model with 24.4% of its parameters pruned across multiple benchmarks. As shown in Table 10, our approach continues to exhibit strong and consistent performance in the PD Unified scenario, demonstrating its robustness even without explicit disaggregation of prefill and decode stages.

## D.4 PERFORMANCE COMPARISON OF DISAGGREGATION AND UNIFIED

To validate the effectiveness of our strategy, we conduct multiple benchmark tests on LLaMA2-13B by removing 25% of the parameters. The comparison method, PD Unified, employs the same parameter removal combination strategy for both prefill and decode stages, without considering the distinct sensitivity of each stage to pruning. As shown in Table 11, the results show that our PD disaggregation compression strategy achieves better performance than PD Unified, which also indicates that our method also performs well under the PD Unified configuration.

Table 9: Evaluation of Combined Pruning and 8-bit AWQ Quantization on Qwen2.5-32B.

| Method | PIQA | Winog | HellaSwag | ARC-E | ARC-C |
|--------|------|-------|-----------|-------|-------|
| Dense | 81.88 | 75.3 | 64.91 | 80.51 | 53.41 |
| FLAP | 77.56 | 61.08 | 55.23 | 70.36 | 42.37 |
| Ours | **78.45** | **72.85** | **57** | **75.8** | **46.67** |
| Ours+Quant | 78.21 | 71.98 | 56.88 | 75.21 | 46.1 |

Table 10: Evaluation of pruning performance in the PD Unified scenario with LLaMA2-13B.

| Method | MMLU | CMMLU | ARC-E | ARC-C | HSwag | MathQA | WNLI | RTE | AVG |
|--------|------|-------|-------|-------|-------|--------|------|-----|-----|
| Dense | 52.10 | 34.73 | 79.42 | 48.38 | 60.07 | 32.09 | 66.20 | 69.31 | 55.29 |
| LLM-Pruner | 50.11 | 33.57 | 61.16 | 37.67 | 47.37 | 24.41 | 43.24 | 59.19 | 44.59 |
| FLAP | 49.89 | 33.91 | 60.90 | 37.65 | 47.54 | 24.78 | 43.26 | 59.19 | 44.64 |
| Shortened | 26.71 | 25.50 | 26.26 | 22.61 | 25.76 | 18.89 | 43.66 | 52.71 | 30.26 |
| ShortGPT | 50.13 | 33.97 | 61.32 | **37.88** | 47.71 | 24.82 | 43.66 | 59.57 | 44.88 |
| SLEB | 23.76 | 25.41 | **67.85** | 33.87 | 48.76 | 25.46 | 45.07 | 58.84 | 41.13 |
| Ours (Unified) | **51.05** | **33.99** | 64.02 | **37.88** | **48.83** | **26.06** | 53.12 | **63.54** | **47.31** |

Table 11: Performance comparison between PD Unified and PD Disaggregation on LLaMA2-13B.

| Strategy | MMLU | CMMLU | ARC-E | ARC-C | HSwag | MathQA | WNLI | RTE | AVG |
|----------|------|-------|-------|-------|-------|--------|------|-----|-----|
| Unified | 51.05 | 33.99 | 64.02 | 37.88 | 48.83 | 26.06 | 53.12 | 63.54 | 47.31 |
| Disaggregation | **51.52** | **35.37** | **69.78** | **40.02** | **53.11** | **28.44** | **53.52** | **64.26** | **49.50** |

# E ABLATION STUDY

We conduct comprehensive ablation studies to evaluate the individual contributions of the two key components of our framework: the iterative block removal strategy and the attention score based filtering mechanism for KV Cache pruning. The experimental results, as illustrated in Figure 2, demonstrate that both components significantly enhance model performance across a diverse range of benchmarks. We conduct experiments on LLaMA3.1-8B, involving the pruning of 9.38% blocks and the application of KV Cache across 22 layers. The other hyperparameter settings used in these experiments are in the implementation details above. Block pruning performance, as shown in Figure 2a, is significantly enhanced by the iterative optimization mechanism, which consistently outperforms its non iterative counterpart. This advantage stems from the iterative strategy's ability to more effectively navigate the search space, ultimately identifying superior block combinations that yield notable improvements in accuracy. The attention score based filtering mechanism in KV Cache pruning, illustrated in Figure 2b, delivers substantial performance gains. It selectively excludes layers with low attention scores, preserving cache integrity in semantically critical regions while efficiently pruning less relevant layers.

# F HYPERPARAMETER IMPACT ANALYSIS

We evaluate the impact of key hyperparameters in our iterative block removal strategy, including the initial temperature $T$, temperature decay coefficient $\alpha$, and minimum temperature $T_{min}$. We prune 9.38% blocks on LLaMA3.1-8B. As shown in Table 12, increasing $T$ and $\alpha$ from the first configuration to the second configuration improves the average accuracy from 63.49 to 63.56. This indicates that a slower cooling schedule helps more effectively explore the solution space and avoid converging to suboptimal local minima. Importantly, further increasing the parameter values in the third configuration does not change the performance across all benchmarks, suggesting that the pruning process has converged. This convergence implies that the algorithm has identified a near optimal block removal combination, beyond which additional iterations or a larger search space will not yield further gains. Overall, all three configurations are able to find a high quality solution space.

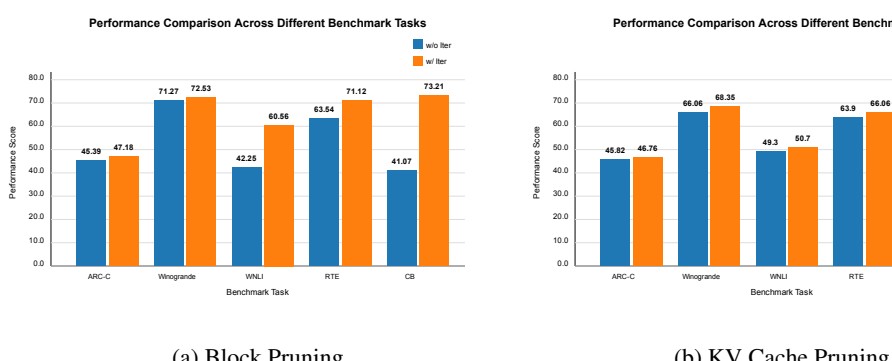

<table>
<tr><td>(a) Block Pruning</td><td>(b) KV Cache Pruning</td></tr>
</table>

Figure 2: Ablation study on the effectiveness of key components: (a) Accuracy comparison with and without the iterative block removal strategy. (b) Accuracy comparison with and without the attention score based filtering mechanism in KV Cache pruning.

Table 12: Performance comparison across three key hyperparameter settings on the LLaMA3.1-8B. Under all these different parameter configurations, the model maintains a high level of performance.

| $T$ | $\alpha$ | $T_{min}$ | ARC-E | ARC-C | PIQA | Winog | BoolQ | MathQA | WNLI | SST-2 | RTE | CB | MMLU | CMMLU | HSwag | AVG |
|---|---|---|---|---|---|---|---|---|---|---|---|---|---|---|---|---|
| 5 | 0.70 | 0.10 | 76.77 | 47.53 | 79.11 | 73.09 | 80.24 | 33.84 | 59.15 | 63.53 | 67.51 | 75.00 | 62.79 | 49.97 | 56.78 | 63.49 |
| 10 | 0.80 | 0.10 | 78.11 | 47.53 | 78.62 | 71.82 | 79.08 | 36.31 | 61.97 | 57.45 | 72.56 | 73.21 | 63.43 | 49.91 | 56.33 | 63.56 |
| 15 | 0.85 | 0.05 | 78.11 | 47.53 | 78.62 | 71.82 | 79.08 | 36.31 | 61.97 | 57.45 | 72.56 | 73.21 | 63.43 | 49.91 | 56.33 | 63.56 |

Additionally, we further assess the threshold parameter $d_T$ for constructing the distillation set. We prune 9.38% blocks of the LLaMA3.1-8B model. As shown in Table 13, the experimental results indicate that different threshold parameters can all achieve relatively optimal outcomes, consistently demonstrating good performance. Additionally, we systematically evaluate the impact of two key hyperparameters in our KV Cache pruning strategy: the retention ratio $p$, which denotes the proportion of tokens preserved from the beginning and end, and the filtering threshold $\gamma$. We prune the KV Cache across 22 layers on LLaMA3.1-8B. As shown in Table 14, the experimental results indicate that various settings of the retention rate $p$ can achieve satisfactory performance. The threshold $\gamma$ determines the intensity of layer filtering. An excessively high threshold ($\gamma \rightarrow 1$) overly restricts the pruning candidates, thereby losing potential opportunities for efficiency improvement. Conversely, an overly lenient threshold ($\gamma \rightarrow 0$) introduces noise from low scoring attention heads. Our scoring metric prioritizes layers with high attention strength and low variance. Regardless of the specific parameter settings, our method consistently demonstrates strong performance across a wide range of configurations.

Table 13: Evaluation of the influence of the block distillation threshold $d_T$ on the LLaMA3.1-8B.

| $d_T$ | ARC-E | ARC-C | PIQA | Winog | BoolQ | MathQA | WNLI | SST-2 | RTE | CB | MMLU | CMMLU | HSwag | AVG |
|---|---|---|---|---|---|---|---|---|---|---|---|---|---|---|
| 0.97 | 77.44 | 47.18 | 78.40 | 72.53 | 76.82 | 36.82 | 60.56 | 59.75 | 71.12 | 73.21 | 58.74 | 48.39 | 55.38 | 62.80 |
| 0.96 | 78.11 | 47.53 | 78.62 | 71.82 | 79.08 | 36.31 | 61.97 | 57.45 | 72.56 | 73.21 | 63.43 | 49.91 | 56.33 | 63.56 |
| 0.95 | 77.65 | 47.18 | 78.40 | 73.40 | 79.57 | 36.21 | 64.79 | 69.72 | 73.29 | 76.79 | 60.76 | 43.37 | 57.22 | 64.49 |
| 0.94 | 76.77 | 47.53 | 79.11 | 73.09 | 80.24 | 33.84 | 59.15 | 63.53 | 67.51 | 75.00 | 62.79 | 49.97 | 56.78 | 63.49 |

## G  PERFORMANCE COMPARISON WITH OTHER KV CACHE PRUNING METHODS

To validate the effectiveness of our approach, we conduct comprehensive performance comparisons against a Dense baseline as well as several recent KV Cache pruning methods, including LazyLLM, MInference and FlexPrefill. Since SlimInfer is not publicly available, it is excluded from our experiments. We evaluate LLaMA3.1-8B across a diverse suite of datasets, including HPQA, 2Wiki, TQA, MuSiQue and SAMSum. This enables a thorough assessment of our method's generality and robustness across different problem domains. To ensure a fair comparison, we maintain a consis-

Table 14: Assessment of the influence of token retention ratio $p$ and attention score filtering threshold $\gamma$ in the KV Cache pruning strategy on the LLaMA3.1-8B model across diverse benchmarks.

| $p$ | $\gamma$ | ARC-E | ARC-C | PIQA | Winog | BoolQ | MathQA | WNLI | SST-2 | RTE | CB | MMLU | CMMLU | HSwag | AVG |
|---|---|---|---|---|---|---|---|---|---|---|---|---|---|---|---|
| 0.10 | 0.40 | 79.88 | 47.61 | 79.60 | 68.19 | 80.49 | 33.00 | 50.70 | 72.59 | 66.79 | 64.29 | 56.94 | 44.47 | 58.15 | 61.75 |
| 0.10 | 0.50 | 79.25 | 48.04 | 79.27 | 69.85 | 80.55 | 31.16 | 47.89 | 74.77 | 68.23 | 66.07 | 55.78 | 42.85 | 58.02 | 61.67 |
| 0.20 | 0.50 | 79.88 | 47.61 | 79.60 | 68.19 | 80.49 | 33.00 | 50.70 | 72.59 | 66.79 | 64.29 | 56.94 | 44.47 | 58.15 | 61.75 |
| 0.20 | 0.60 | 78.99 | 48.04 | 78.89 | 70.09 | 81.04 | 31.36 | 54.93 | 70.76 | 70.76 | 64.29 | 57.54 | 44.94 | 58.07 | 62.28 |
| 0.30 | 0.80 | 78.75 | 46.76 | 79.16 | 68.35 | 78.59 | 30.65 | 50.70 | 74.77 | 66.06 | 64.29 | 54.09 | 40.36 | 57.35 | 60.76 |

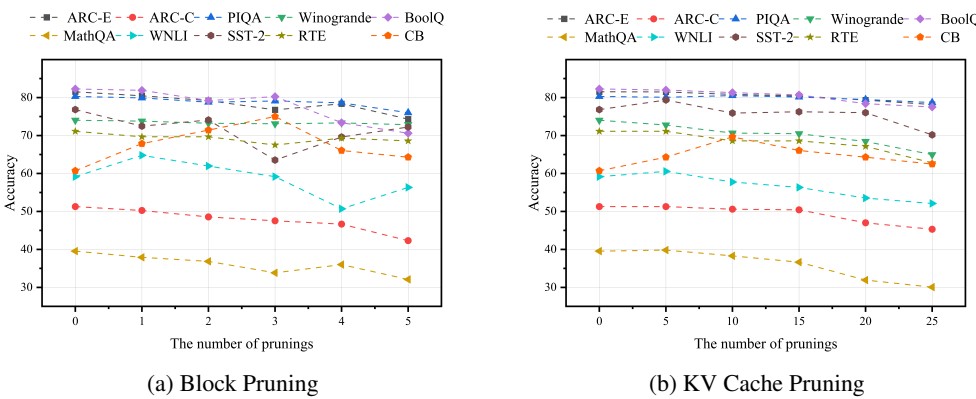

(a) Block Pruning          (b) KV Cache Pruning

Figure 3: Sensitivity analysis of model accuracy under varying pruning ratios in LLaMA3.1-8B.

tent token pruning ratio of 0.5 across all methods, and all hyperparameters are set according to the configurations reported in the original works. As shown in Table 15, our method consistently outperforms other approaches, demonstrating its effectiveness in preserving inference quality. These results underscore the potential of our method as a practical and efficient solution for LLM deployment.

Table 15: Performance comparison with other KV Cache pruning methods.

| Method | HPQA | 2WiKi | TQA | MuSiQue | SAMSum | AVG |
|---|---|---|---|---|---|---|
| Dense | 55.50 | 44.28 | 91.65 | 30.78 | 43.92 | 53.23 |
| LazyLLM | 54.52 | 43.42 | 91.00 | 28.86 | 43.64 | 52.29 |
| MInference | 52.00 | 44.10 | 91.18 | 25.72 | 43.73 | 51.35 |
| FlexPrefill | 54.56 | 43.43 | 89.81 | 30.07 | 43.18 | 52.21 |
| **Ours** | **55.08** | **44.16** | **91.27** | **30.25** | **43.81** | **52.91** |

## H SENSITIVITY ANALYSIS OF VARYING PRUNING RATIOS

Figure 3 illustrates the impact of different pruning ratios on LLaMA3.1-8B performance, covering both block pruning and KV Cache pruning scenarios. As the pruning ratio increases, the accuracy of most tasks remains relatively stable despite a general downward trend, indicating the strong robustness of our method. Block pruning accuracy remains stable even when the number of pruned blocks increases, especially on tasks such as PIQA, Winogrande and RTE. This phenomenon validates the effectiveness of our redundancy aware iterative block pruning strategy, which selectively removes non-critical blocks while preserving the model's core functional capabilities. Our results for KV Cache pruning demonstrate even stronger robustness. As shown in Figure 3b, most tasks maintain high accuracy even after pruning. In practical scenarios, the pruning ratio can be adjusted to balance the trade-offs according to specific requirements. This gradual decline in performance highlights the graceful degradation characteristic of this method, underscoring its practical value in memory constrained inference scenarios.

# I    ANALYSIS OF USING COSINE SIMILARITY

Besides cosine similarity, dot product and Euclidean distance are also commonly used, but they are sensitive to vector magnitude. Previous work (Chen et al., 2024) has shown that in Transformers, hidden states tend to expand with depth, leading deeper layers to exhibit higher dot-product similarity, while shallower layers maintain smaller Euclidean distances. To avoid this bias, we use cosine similarity, which is insensitive to magnitude. In addition, we measure the L2 norms of the hidden states at each layer of Qwen2.5-7B. As shown in Table 16, the norm magnitude exhibits a clear upward trend with increasing depth, which highlights the inherent bias of other similarity metrics that are sensitive to vector magnitude. Furthermore, we evaluate the performance of Qwen2.5-7B and OPT-13B under 15% parameter pruning with different similarity metrics. As shown in Table 17, cosine similarity consistently outperforms the other metrics.

Table 16: The norms of the hidden states output at each layer of Qwen2.5-7B. The first row represents the layer IDs, and the second row represents the corresponding norms.

| **Layer** | 0 | 1 | 2 | 3 | 4 | 5 | $\cdots$ | 21 | 22 | 23 | 24 | 25 | 26 | 27 |
|---|---|---|---|---|---|---|---|---|---|---|---|---|---|---|
| **Norm** | 14 | 19 | 25 | 116 | 141 | 146 | $\cdots$ | 264 | 300 | 350 | 408 | 470 | 474 | 620 |

Table 17: Performance comparison on Qwen2.5-7B using different similarity metrics.

| **Metric** | **PIQA** | **Winog** | **HSwag** |
|---|---|---|---|
| Euclidean distance | 75.70 | 52.20 | 50.51 |
| Dot product | 75.87 | 55.88 | 47.62 |
| Cosine similarity | 77.04 | 56.99 | 51.19 |

# J    MORE ANALYSIS

## J.1    ROBUSTNESS ON DIFFERENT CALIBRATION SETS

To evaluate the stability of our proposed block pruning strategy, we systematically analyze the consistency of results across different calibration sets and sizes. Our analysis distinguishes between two types of removal operations: direct block pruning and block distillation. Direct block pruning refers to the removal of blocks, while block distillation involves merging blocks instead of removing them. As shown in Table 18, the experimental results demonstrate significant stability in the selection of removed blocks across all experimental conditions. This consistency strongly indicates that our pruning decisions are driven by the inherent architectural redundancy within the model itself, exhibiting robustness across different datasets.

Table 18: Results comparison across different calibration sets. Under the LLaMA3.1-8B model, we report the selected block indices for removal across different calibration datasets and various sizes under two pruning configurations: removing 15.63% blocks and removing 9.38% blocks. The indices within square brackets denote the distilled blocks.

| **Calibration Set** | **Size** | **15.63% Blocks Removal** | **9.38% Blocks Removal** |
|---|---|---|---|
| PIQA | 256 512 1024 | 24, 25, 26, [22, 23], [27, 28] | 25, 26, [27, 28] |
| WikiText-2 | 256 512 1024 | 24, 25, 26, [22, 23], [27, 28] | 25, 26, [27, 28] |

Furthermore, we validate the robustness of our KV Cache pruning method across different datasets. Specifically, we use the KV Cache pruning layers tested on TQA as a reference and compare them with the results obtained on PIQA and WikiText-2. We measure the similarity between the results using the Jaccard coefficient, which quantifies the similarity between two sets as the size of their intersection divided by the size of their union. A higher coefficient indicates greater similarity, while a lower value indicates less similarity. As shown in Table 19, the Jaccard coefficient across datasets is 1, demonstrating that our method is robust with respect to dataset variations.

Table 19: Robustness of KV Cache pruning across datasets measured by the Jaccard coefficient.

| Dataset | Jaccard Coefficient |
|---|---|
| TQA (reference) | 1.0 |
| PIQA | 1.0 |
| WikiText-2 | 1.0 |

### J.2 CALIBRATION LATENCY (PRUNING AND DISTILLATION SETS CONSTRUCTION & KV CACHE PRUNING)

To further demonstrate the efficiency advantages of our method, we provide a detailed latency analysis of the calibration process. This process mainly involves the construction of the initial pruning set and distillation set, together with the KV Cache pruning applied to the selected layers. We carry out experiments on both LLaMA3.1-8B and LLaMA2-13B, with pruning ratios of 15.63% and 24.4%, respectively. We further evaluate our method by comparing its pruning latency against the KV Cache pruning approach LazyLLM. As summarized in Table 20, we report the time required for pruning and distillation sets construction, as well as the latency introduced by KV Cache pruning. The results indicate that these operations can be completed within a very short time, resulting in only negligible overhead and imposing virtually no additional burden.

Table 20: Time for constructing pruning and distillation sets, and KV cache pruning during the calibration process (in seconds) on LLaMA series.

| Model | Method | Construction | KV Cache |
|---|---|---|---|
| LLaMA3.1-8B | LazyLLM | - | 26.18 |
| | Ours | 4.23 | 4.51 |
| LLaMA2-13B | LazyLLM | - | 53.16 |
| | Ours | 9.43 | 9.76 |

## K ANALYSIS OF ITERATIVE BLOCK REMOVAL SOLUTIONS

To validate the effectiveness of our iterative block removal method, we compare its solutions with the global optimum obtained via exhaustive search. It is important to note that exhaustive search requires evaluating all possible block combinations, which is practically intractable. To make this comparison feasible, we select all elements from the distillation set and the top five elements from the pruning set, and treat the optimal solution among all their combinations as the global optimum. Even so, solving it still demands substantial computational time. We apply our method to prune 13.6% of the parameters in LLaMA3.1-8B and evaluate it across various benchmarks. As shown in Table 21, our approach achieves results consistent with the global optimum. This demonstrates that our iterative method efficiently finds superior pruning solutions.

Table 21: Performance of iterative block removal compared to the global optimum on LLaMA3.1-8B.

| Method | MMLU | CMMLU | ARC-E | ARC-C | PIQA | Winog | HSwag | BoolQ | MathQA |
|---|---|---|---|---|---|---|---|---|---|
| global optimum | 60.82 | 46.66 | 73.15 | 43.86 | 76.99 | 73.56 | 53.80 | 69.85 | 34.28 |
| Ours | 60.82 | 46.66 | 73.15 | 43.86 | 76.99 | 73.56 | 53.80 | 69.85 | 34.28 |

## L    COMPARISON WITH WORK OF SIMILAR NAMES

We compare our approach with another method (Muralidharan et al., 2024) that bears a similar name, as both involve concepts such as iteration and distillation. In their work, iteration refers to repeatedly performing pruning to gradually reduce the model size, whereas in our approach, iteration is used to search for the optimal pruning configuration, and the two notions are not directly related. Similarly, their distillation is applied after pruning to the entire model in order to restore performance, while our distillation occurs during the pruning process itself, serving as a mechanism for block removal.

## M    MORE DETAILS ON PREFILL AND DECODE REMOVAL SCHEMES

Our approach first determines the optimal removal strategy and then derives separate removal schemes for the prefill and decode stages. Specifically, the prefill scheme is designed as a subset of the decode scheme, ensuring that every retained block in the decode node can directly reuse the corresponding KV Cache from the prefill node. In contrast, if separate removal schemes are determined for the prefill and decode stages at the initial stage, without any subset relationship, it is likely that some blocks retained in the decode node would have no corresponding KV Cache in the prefill node, hindering cache reuse and reducing system efficiency.

## N    CASE STUDY

To qualitatively assess the impact of pruning on generation quality, we compared the outputs of the original model and the pruned model for several representative prompts, as shown in Table 22. We prune 9.38% blocks on LLaMA3.1-8B. Despite a substantial reduction in model size, the pruned model consistently generates coherent, information-rich, and well-structured responses, retaining the core semantics of the original output. These examples confirm that our pruning strategy maintains strong generation capabilities and factual consistency even under significant structural compression.

## O    LIMITATIONS AND FUTURE WORK

In this work, we primarily focus on the fundamental model of PD disaggregation. However, we do not fully consider some of the more complex management aspects of pruning in PD disaggregation, such as memory management. Under the PD disaggregation framework, dynamically constructing pruned subgraphs introduces considerable memory management overhead. To address this challenge, we plan to propose pruning-aware memory management strategies. One approach involves statically preallocating memory blocks for frequently observed subgraph patterns, such as common structural pruning configurations to avoid repetitive allocation overhead. We also apply tensor lifetime analysis to enable memory reuse across subgraphs. Furthermore, we can separate structural masks (such as sparsity metadata) from computation tensors to reduce memory pressure. Moreover, we can explore MoE pruning within the PD disaggregation framework by removing infrequently used expert subgraphs during the prefill phase and dynamically reusing the pruned subgraphs based on context in the decode phase. Additionally, leveraging tensor lifetime analysis enables memory reuse across MoE subgraphs, further improving efficiency.

## P    THE USE OF LARGE LANGUAGE MODELS (LLMS)

Large language models (LLMs) are employed in this work exclusively for the purpose of polishing the written presentation, such as improving grammar, clarity and readability. The research ideas, technical framework, theoretical analyses, experimental design and conclusions are fully conceived and carried out by the authors. The role of the LLM is limited to enhancing the fluency and style of the manuscript without influencing the originality, novelty or scientific content of the work.

Table 22: Qualitative examples comparing outputs from the original and the pruned versions of the LLaMA3.1-8B models configured as described above. Each example illustrates the model's response to the same input prompt, with the underlined text indicating the user's question.

| Original Model | Pruned Model |
|---|---|
| Explain what artificial intelligence is. Artificial intelligence (AI) is the simulation of human intelligence processes by machines, especially computer systems. These processes include learning (the acquisition of information and rules for using the information), reasoning (using the rules to reach approximate or definite conclusions), and self-correction. Particular applications of AI ... | Explain what artificial intelligence is. Artificial intelligence, or AI, is a field of computer science that deals with the creation of machines that can perform tasks that normally require human intelligence. This includes things such as understanding natural language, recognizing objects in images, and making decisions based on complex data. While AI has been around for decades, recent ... |
| Provide some tips for learning a new language. 1. Set realistic goals: Start by setting realistic goals for yourself, such as learning a few basic phrases or mastering a specific skill. 2. Practice regularly: Consistency is key when it comes to learning a new language. 3. Find a language partner: Finding a language partner can be a great way to practice and improve your language skills. 4. Use technology... | Provide some tips for learning a new language. Here are some tips for learning a new language: 1. Start by learning the basics of the language, such as the alphabet, common words and phrases, and basic grammar rules. 2. Find a language learning app or program that suits your learning style and goals. 3. Practice speaking and listening to the language as much as possible. 4. Practice ... |
| Explain how a car engine works. A car engine is a machine that converts the chemical energy stored in gasoline into mechanical energy that can be used to move the car. The engine consists of several parts that work together to produce this energy. The first part of the engine is the combustion chamber. This is where the gasoline is mixed with air and ignited to create an explosion. The explosion pushes down on a piston, which is connected to a crankshaft. The crankshaft is what turns wheels ... | Explain how a car engine works. A car engine works by converting chemical energy stored in fuel into mechanical energy that drives the car. The process of converting chemical energy into mechanical energy is known as combustion. In a car engine, combustion occurs in the combustion chamber, which is located in the center of the engine. The combustion chamber is connected to the intake manifold, which is where the fuel and air mixture enters the engine. The intake manifold ... |

