# OpenReview forum: "PDTrim: Targeted Pruning for Prefill-Decode Disaggregation in Inference"
_ICLR.cc/2026/Conference — ICLR 2026 Conference Withdrawn Submission_

### Official Review · Reviewer_LQRH · 2025-10-25

**Soundness:** 2
**Presentation:** 1
**Contribution:** 1
**Rating:** 2
**Confidence:** 4

**Summary:**

This paper proposes a pruning method tailored for prefill-decode (PD) disaggregated inference in large language models. It prunes model blocks and KV caches separately for prefill and decode stages, improving efficiency and accuracy.

**Strengths:**

Significance
1. The paper conducts extensive experiments demonstrating that the proposed method can outperform state-of-the-art approaches.

**Weaknesses:**

Originality

1. The proposed approach is not methodologically novel. Iterative block pruning, cosine similarity scoring, and heuristic search strategies have already been explored in prior works.
2. The paper fails to clearly justify why PD disaggregation requires a special pruning framework. Similar methods can be applied to both unified and disaggregated inference without modification.
3. The KV cache pruning mechanism is largely heuristic and similar to the Sink method, offering limited conceptual advancement.

Quality

4. The method is highly heuristic, relying on many stochastic thresholds and hyperparameters, making results sensitive to tuning and hard to reproduce.
5. The claim of global optimal block pruning is not supported. The paper merely proposes an approximate heuristic search method that lacks theoretical justification.
6. The theoretical part is weak and lacks rigor, with unclear use of Lipschitz assumptions and non-standard theorem formatting. It appears poorly reasoned and seems to have been generated with the assistance of an LLM.
7. Important KV cache pruning baselines such as Sink and H2O are missing, making the comparison incomplete and weakening the motivation for the proposed approach [1,2].
8. The proposed method relies on distillation while several baselines do not, and important block pruning methods such as Wanda-SP are also omitted.
9. The pruning ratios appear carefully tuned, suggesting possible cherry-picking that may bias results.

Clarity

10. Key concepts are unclear or confusing, such as what is meant by the “first and last token sequences” and how they apply in the prefill stage.
11. The paper does not describe how the overhead of KV cache pruning is measured or how attention heads with low scores are handled. The section on KV cache pruning is unclear and lacks sufficient explanation of key design choices.

Significance

12. Despite extensive experimentation, the paper provides limited methodological insight and the gains are incremental and largely empirical.
13. Reported latency improvements are negligible in practice, and the results do not convincingly demonstrate that PD-specific pruning is necessary.
14. While the paper presents extensive experiments, it prioritizes quantity over conceptual clarity, offering very limited methodological insight beyond empirical validation.


[1] Efficient Streaming Language Models with Attention Sinks
[2] H_2O: Heavy-Hitter Oracle for Efficient Generative Inference of Large Language Models

**Questions:**

1. While the paper presents extensive experiments, good research is not about stacking experiments but about providing clear methodological insight. The proposed method lacks a strong conceptual contribution beyond empirical validation, which limits its novelty and broader significance.
2. The motivation for positioning this method specifically for PD disaggregation is not well justified. In principle, the same approach could be applied to both unified and disaggregated PD settings, and other existing methods can also handle such cases, suggesting that nothing in the proposed design is inherently PD-specific.
3. The theoretical analysis lacks rigor and proper formatting, particularly in the theorem sections, which appear superficial and sometimes unsubstantiated, resembling LLM-generated text. A more careful and formal presentation is needed.
4. Add ablation studies (e.g., without distillation, or isolating block vs. cache pruning) to ensure fair and transparent evaluation.
5. Include missing baselines such as Wanda-SP, and ensure fair comparisons with calibration-only methods that do not rely on distillation.
6. Reduce the number of hyperparameters and heuristic thresholds to improve reproducibility and reduce tuning bias.
7. What exactly does “first and last token sequences” mean in the proposed pruning mechanism? The formulation in Equation (19) appears nearly identical to the Sink method [1], suggesting limited novelty.
8.Please avoid claiming global optimal pruning, as the proposed approach is a heuristic approximation to a large-scale search problem rather than a proven optimal solution.
9. How is the overhead of KV cache pruning measured in practice? Existing works such as Sink and H2O [2] provide clearer, more straightforward metrics for such evaluation.

[1] Efficient Streaming Language Models with Attention Sinks
[2] H_2O: Heavy-Hitter Oracle for Efficient Generative Inference of Large Language Models

---

> ### Author Response · Authors · 2025-11-18
>
> Thanks for your detailed review and constructive feedback on our manuscript. We have carefully reviewed all your comments and addressed each concern with clear clarifications and supporting evidence. Please see our detailed responses below.
>
> **R1. Novelty (W1).**
>
> Cosine similarity is not our core contribution. Our work is the first to apply this heuristic iterative strategy to block pruning, and the pruning system specifically designed for PD disaggregation is also the first of its kind.
>
> **R2. Disaggregated Design (W2, Q2).**
>
> Under a disaggregated PD setup, where prefill and decode are deployed on different nodes, two issues arise: (1) the execution processes on different nodes differ, leading to varying error accumulation and sensitivity; (2) there is bandwidth overhead for data transmission. Previous weight selection methods were designed uniformly for both stages, ignoring their distinct characteristics. Additionally, prior KV Cache pruning methods incur high load. All of these fail to meet the high-throughput requirements of a disaggregated PD system.
>
> **R3. KV Cache pruning (W3, W7, W11, Q7, Q8).**
>
> Our KV Cache pruning is designed at the granularity of attention heads and requires only a single calibration process, resulting in very low load and ensuring the throughput of a disaggregated PD system. Dynamic management approaches like Sink introduce additional overhead and fail to meet the high-throughput requirements of disaggregated PD. Experiments in **Appendices J.1, J.2, and G** further demonstrate that our KV Cache pruning outperforms existing methods in terms of ​**accuracy, overhead, and robustness**​.
>
> **In Appendix G,** we compare our method with a large number of recent KV Cache pruning baselines. These new baselines have already been shown to perform better than Sink and H2O. ​**In Appendix J.2**​, we illustrate the overhead.
>
> If a layer contains heads with very low attention scores on the first and last tokens of the sequence, we do not perform KV Cache pruning on that layer​**​ (lines 274–280)**​.
>
> **R4. Hyperparameter (W4, Q6).**
>
> We conducted hyperparameter analysis experiments in ​**Appendix F**​, and our method achieves good performance across different hyperparameter settings.
>
> **R5. Approximate Optimal Solution (W5, Q8).**
>
> Compared to previous greedy methods, our method provides an approximate optimal solution, as truly optimal solutions are often difficult to achieve or prove. Additionally, we include related experiments and analyses in ​**Appendix K**​.
>
> **R6. Theoretical part (W6, Q3).**
>
> Equation 36 is a well-established theoretical result, and in **Appendix B** we also provide a derivation based on Lipschitz continuity.
>
> **R7. Ablation study without distillation / Comparison with Wanda-SP  (W8, Q4, Q5).**
>
> We conducted an ablation study on LLaMA2-13B without distillation, as well as a comparative experiment with Wanda-SP. The results are as follows:
>
> | Method                 | MMLU  | CMMLU | ARC-E | PIQA  |
> | ------------------------ | ------- | ------- | ------- | ------- |
> | Dense                  | 52.10 | 34.73 | 79.42 | 79.05 |
> | Wanda-SP               | 50.13 | 34.06 | 61.26 | 71.11 |
> | Ours (no distillation) | 50.86 | 35.12 | 67.53 | 72.38 |
> | Ours                   | 51.52 | 35.37 | 69.78 | 74.43 |
>
> **R8. Pruning ratio (W9).**
>
> Our pruning is performed at the granularity of entire blocks, and the pruning rates are discrete and finite rather than continuous or unlimited.
>
> **R9. Explanation of first/last (W10, Q7).**
>
> For an input sequence to an LLM, which consists of many tokens, the first and last tokens refer specifically to the several tokens at the very beginning and very end of the sequence. Prefill nodes only transmit the KV Cache corresponding to these tokens at the sequence’s front and back to the decode nodes.
>
> **R10. Latency analysis (W13).**
>
> The reduction in latency is related to the pruning rate; the higher the pruning rate, the greater the latency reduction. In Table 3, we report latency tests at the same pruning rates for our method and other methods. Compared to others, our approach already achieves strong performance in terms of latency.

---

> > ### Comment · Reviewer_LQRH · 2025-11-23
> > **Response**
> >
> > 1. The heuristic iterative strategy for block pruning has been extensively explored in the pruning literature. Fundamentally, the proposed approach functions as a heuristic search method for block pruning. The PD disaggregation component is also not a fundamental methodological advance, but rather an engineering choice arising from hardware limitations.
> > 2. Demonstrating KV-cache improvements experimentally does not fully address my concern that the proposed method remains conceptually similar to Sink and offers limited genuine novelty.
> > 3. The theoretical analysis is weak and appears superficial. It resembles patterns commonly generated by LLMs, particularly the reliance on Lipschitz-style arguments to circumvent deeper theoretical challenges. Such loose bounding does not provide meaningful insight into the method’s behavior.
> > 4. I would like to reiterate that simply showing improvements across many experiments does not by itself make the research strong, because it provides little insight that others can learn from. Without real conceptual or methodological innovation, the work becomes just another method to compare against, rather than one that deepens our understanding.
> >
> > Given that most of these concerns remain insufficiently addressed, I have decided to maintain my original evaluation.

---

### Official Review · Reviewer_goEM · 2025-11-01

**Soundness:** 2
**Presentation:** 2
**Contribution:** 3
**Rating:** 4
**Confidence:** 3

**Summary:**

PDTrim targets real-world prefill–decode (PD) disaggregation, where prefill and decode run on different resources. The method (1) builds separate pruning/distillation sets and performs iterative block pruning independently for prefill vs. decode; (2) introduces a KV-cache pruning rule that selectively keeps entries (e.g., first/last segments) for designated layers to cut bandwidth. Experiments show better accuracy/latency trade-offs in both disaggregated and unified settings, and up to $4.95\times$ reduction in data transmission.

**Strengths:**

1. Treats bandwidth as a first‑class objective in pruning for PD.

2. Demonstrates improvements across several backbones and settings.

3. Compatible with quantization methods, suggesting composability.

**Weaknesses:**

1. Heuristic criteria: attention‑score cutoffs and first/last‑segment retention may fail on datasets with mid‑context dependencies (e.g., retrieval‑augmented inputs).

2. Limited scaling evidence: no systematic results for long contexts (e.g., $>$64k tokens) or for many concurrent decoders.

3. Search overhead & stability: iterative pruning/annealing costs and variance across runs are insufficiently reported.

4. Comparative breadth: needs head‑to‑head comparisons with recent KV compression, speculative decoding, and structured pruning under matched budgets.

**Questions:**

1. How robust is the KV pruning policy for tasks with mid‑sequence salient tokens (long‑form QA, multi‑turn chat)?

2. What is the wall‑clock cost and variance of the iterative block search for 13B/32B models?

3. Can pruning decisions be made online using gradients or saliency estimates gathered during serving?

4. How does PDTrim interact with speculative decoding and KV quantization?

---

> ### Author Response · Authors · 2025-11-18
>
> Thanks for your detailed review and constructive feedback on our manuscript. We have carefully reviewed all your comments and addressed each concern with clear clarifications and supporting evidence. Please see our detailed responses below.
>
> **R1. KV Cache pruning (W1, W2 and Q1).**
>
> The fact that mid-sequence tokens often have low attention scores in the KV cache has already been demonstrated in several works, and is consistently observed across a wide range of tasks [1]. These intermediate tokens typically receive near-zero attention and can thus be safely ignored [1]. In essence, during the prefill stage, the attention mechanism enables the KV cache produced by later tokens to partially retain the information carried by the intermediate tokens. Compared with these methods, our approach introduces a lower load. **Appendix J.1** demonstrates the robustness of our KV cache method. ​**In Appendix G**​, we further evaluate tasks involving very long sequences and tasks with substantial mid-sequence information. Our method consistently maintains strong performance.
>
> [1] H\_2O: Heavy-Hitter Oracle for Efficient Generative Inference of Large Language Models
>
> **R2. Orthogonal to other methods (W2 and Q4).**
>
> Our method does not significantly modify the LLM architecture, which still consists of the three components: embedding, transformer layers, and heads. Multiple heads can still be attached to the final layer for speculative decoding without any issues. Likewise, the KV cache can still be quantized, allowing both techniques to be used simultaneously. Below, we present results on Qwen2.5-32B demonstrating the combination with KV cache quantization.
>
> | Method        | PIQA  | Winog | HellaSwag | ARC-E |
> | --------------- | ------- | ------- | ----------- | ------- |
> | Dense         | 81.88 | 75.3  | 64.91     | 80.51 |
> | Ours          | 78.45 | 72.85 | 57        | 75.8  |
> | Ours+KV-Quant | 78.28 | 72.31 | 56.9      | 75.8  |
>
> **R3. Execution cost (W3 and Q2).**
>
> In ​**Table 5 (lines 421–434)**​, we report the execution time costs for the 8B and 13B models and compare them with SLEB, showing that our method achieves higher accuracy while incurring lower execution costs. We have additionally tested the 32B model, which requires 251.08 seconds of execution time, which is still less than SLEB’s execution time for the 13B model.
>
> **R4. Comparative experiments (W4).**
>
> Under the same budget, we compare our method with structured pruning approaches in ​**Tables 1 and 2**​, achieving strong results. **In Appendix G, ​**we also compare with recent KV cache pruning methods and still observe consistently good performance.
>
> **R5. Method applicability (Q3).**
>
> We can make pruning decisions using collected gradients or saliency estimates, as these are merely different ways of measuring weight importance. **Lines 361–368** also demonstrate that our method can be fully applied using alternative metrics.

---

### Official Review · Reviewer_brgU · 2025-11-01

**Soundness:** 2
**Presentation:** 3
**Contribution:** 3
**Rating:** 4
**Confidence:** 3

**Summary:**

This paper introduces PDTrim, a pruning framework designed specifically for Prefill-Deocde disaggregation in large language model inference. While conventional pruning methods treat model layers uniformly, PDTrim recognizes that the prefill and decode stages have heterogeneous resource chracteristics and pruning sensitivities. The proposed framework combines two complementary mechanisms:


(1) stage-aware block pruning: Iteratively removes redundant transformer blocks for each stage, guided by cosine similarity and probabilistic optimization.

(2) Selective  KV cache pruning: retains cache entries for only the most attended tokens, reducing communication bandwith between prefill and decode nodes.

Experiments across multiple LLMs dmonstrate competitive performance compared with baselines such as LLM-Pruner, ShortGPT, SLEB and Shorten LLaMA, achieving up to 4.95x reduction in bandwidth and faster inference with comparable accuracy.

**Strengths:**

(1) The paper reports results on diverse model scales and benchmark suites, including MMLU, HellaSwag, ARC, BoolQ and others.

(2) Demonstrates consistent gains over strong baselines, with signifcant latency and bandwidth improvements.

(3) The work connects algorithmic pruning design with practical deployment efficiency.

**Weaknesses:**

(1) Several tables omit key baselines or benchmarks without clear explanation. Formatting issues reduce credibility of reported outcomes.

(2) Unclear technical formulation
(2-1) The construction of pruning and distillation sets in Section 2.1.1 assigns every block to one of the two sets, yet introduces a threshold $d_{T}$ whos operational role is ambiguous.
(2-2) It remains unclear how blocks below the threshold are handled and how these sets interact during iterative optimization.

(3) The paper reports LLM-Pruner and Short GPT variants but lacks sufficient detail on how PDTrim was incorporated into those framework.

(4) In some unified (non-PD) settings, PDTrim even outperforms the PD-Disaggregated case, which weakens the claim that its benefits stem from PD awareness itself.

(5) Dense mathematical expressions and inconsistent figure-text correspondence make the method challenging to follow.

**Questions:**

(1) In several cases, PDTrim performs better in unified (non-PD) inference than in PD-disaggregated setups. Could you clarify whether the performance gains are due to PD disaggregation or just from the pruning heuristics themselves?

(2) Some benchmarks and baselines (e.g., ARC-E, HellaSwag, SLEB) are inconsistently included across tables. Is there a specific reason for their exclusion?

(3) The pruning formulation combines cosine similarity, distillation pairing, and simulated annealing. Was this derived from prior theoretical work or developed empirically?

(4) Can the authors provide ablation results isolating the contribution of Stage-Aware Pruning vs KV Cache Pruning? This would clarify which part contributes most to bandwidth reduction.

---

> ### Author Response · Authors · 2025-11-18
>
> Thanks for your detailed review and constructive feedback on our manuscript. We have carefully reviewed all your comments and addressed each concern with clear clarifications and supporting evidence. Please see our detailed responses below.
>
> **R1. Comparison Results (W1 and Q2).**
>
> Although Table 2 contains fewer benchmarks than Table 1, it still covers a very large number of benchmarks compared to previous related pruning work, including a wide range of tasks, making it sufficiently comprehensive. Adding more data would result in very small font sizes due to page width constraints. We add additional benchmark data for LLaMA3.1-8B below.
>
> | Method           | PIQA  | Winog | CB    |
> | ------------------ | ------- | ------- | ------- |
> | LLM-Pruner       | 74.54 | 69.02 | 55.28 |
> | ShortGPT         | 74.48 | 69.38 | 55.36 |
> | LLM-Pruner(Ours) | 76.12 | 72.81 | 68.93 |
> | ShortGPT(Ours)   | 75.99 | 72.85 | 65.62 |
> | Ours             | 76.96 | 73.56 | 73.21 |
>
> Taking Table 9 as an example, since previous experiments have already compared a large number of baselines, we select the best-performing baselines for comparison here. We further supplement the comparison with the SLEB and LLM-Pruner baselines as follows.
>
> | Method     | PIQA  | Winog | HSwag | ARC-E | ARC-C |
> | ------------ | ------- | ------- | ------- | ------- | ------- |
> | Dense      | 81.88 | 75.3  | 64.91 | 80.51 | 53.41 |
> | LLM-Pruner | 78.02 | 61.64 | 55.02 | 70.24 | 42.24 |
> | SLEB       | 72.28 | 65.88 | 56.32 | 69.22 | 38.61 |
> | Ours       | 78.45 | 72.85 | 57    | 75.8  | 46.67 |
>
> **R2. Technical Scheme (W2).**
>
> Each element in the distillation set consists of two consecutive blocks, while each element in the pruning set consists of a single block. For a pair of consecutive blocks, if they satisfy the threshold **$d\_T$**, the pair is included in the distillation set; otherwise (i.e., below the threshold), both blocks are included in the pruning set. In both the distillation and pruning sets, each element corresponds to the removal of a single block. During the iterative optimization process, we select a suitable combination of elements from the distillation and pruning sets to determine which blocks to prune (search for an approximate global optimum among all possible combinations).
>
> **R3. Clarification on Variants (W3).**
>
> LLM-Pruner and ShortGPT each provide their own methods for selecting pruning weights. We use their approaches to choose the pruning weights for the decode stage, while for the prefill stage, we apply our own method for selecting pruning weights (as described in lines 245–252). In short, the pruning weights for prefill are a subset of the pruning weights for decode. This way, the pruning results for the two stages are determined separately using different selection strategies.
>
> **R4. Comparison of Disaggregated and Unified Results (W4 and Q1).**
>
> Thank you for your suggestion. We compared the results in Tables 1, 2, 10, and 11, and did not observe the phenomenon you mentioned. Moreover, even if such cases occur for a few data points, this is reasonable, as it is a common phenomenon in pruning during benchmark evaluations. In Tables 10 and 11, we present comparative experiments of various methods under the non-PD disaggregated setting, as well as comparisons between PD Disaggregated and PD Unified, all demonstrating their effectiveness.
>
> **R5. Explanation of the Figures and Formulas (W5).**
>
> We provide the detailed theoretical proofs in the appendix, while the main text only presents the conclusions. The theory demonstrates the differing sensitivities of prefill and decode. Readers interested in the derivation can refer to the appendix, while those less concerned with the detailed reasoning can simply note that prefill is more sensitive than decode.
>
> Figure 1 provides an overview of our method. The upper part illustrates the iterative process over the distillation and pruning sets to obtain the final pruning solution. The lower part shows the differences between the prefill and decode pruning blocks, and indicates that, during KV cache transmission, only the first and last portions are sent for certain layers.
>
> **R6. Origin of the Method (Q3).**
>
> Our method builds partially on prior work; for example, ShortGPT measures block redundancy using cosine similarity (which is not our core contribution). More importantly, we analyzed the shortcomings of existing algorithms, proposed potential solutions from a theoretical perspective, and then validated their effectiveness through experiments.

---

> ### Author Response · Authors · 2025-11-18
>
> **R7. Contribution analysis (Q4).**
>
> The contribution of each component depends on the pruning parameters. To more clearly illustrate their respective effects, we provide a theoretical explanation. Our stage-aware block pruning and KV-cache pruning both reduce data transfer, thereby lowering bandwidth usage. Suppose a layer requires transferring an amount of data denoted as **$n$**. Pruning one block reduces the layer’s transfer cost to zero, contributing **$n$**. If the first–last pruning ratio for the KV cache is $p$ (where $p$*​*​ < 0.5​), then the reduced data transfer is 2 * n * p. If **$a_1$** layers undergo stage-aware block pruning and **$a_2$** layers apply KV-cache pruning, their contributions to bandwidth reduction are as follows:
>
> | Method                    | Contribution              |
> | --------------------------- | --------------------------- |
> | stage-aware block pruning | **$$a_1*n$$**       |
> | KV-cache pruning          | **$a_2$  * 2 * n * p** |

---

### Official Review · Reviewer_3b4H · 2025-11-03

**Soundness:** 3
**Presentation:** 3
**Contribution:** 3
**Rating:** 6
**Confidence:** 4

**Summary:**

The paper proposes a pruning method PDTRIM, targeting for prefill–decode (PD) disaggregated inference. The framework learns separate block-removal plans for prefill vs. decode and prunes KV-cache transfers from prefill to decode. It builds pruning and distillation sets, defines block redundancy via cosine similarity between a block’s input/output, and searches the space of block removals with a simulated-annealing style iteration to find stage-specific solutions. To reduce inter-node bandwidth, it selects layers whose attention strongly concentrates on the first/last tokens and, for those layers, reuses only the first/last token KV entries from prefill while keeping decode-side KV intact. Experiments show faster inference and up to 4.95× less data transmitted, with accuracy competitive or better than strong pruning baselines.

**Strengths:**

1. The proposed method learns separate pruning plans for prefill and decode, which is different from prior pruning methods that mainly focus on the prefill stage.. The paper explicitly states this and analyzes why prefill is more sensitive (errors accumulate across generated tokens). Reported results show lower inference latency versus other baselines and better accuracy under PD disaggregation than unified one-stage pruning.

2. Bandwidth-aware KV pruning matches the real bottleneck in PD process. The first/last-token reuse rule is simple, tunable, and empirically effective, with concrete bandwidth savings.

3. The algorithm descriptions, practical hyperparameters are clearly shown.  And the paper reports broad evaluation across models and tasks, including direct comparisons to recent KV-pruning methods under a common token-pruning ratio.

4. Ablation shows that the iterative search and the KV-layer selection both contribute to the framework.

**Weaknesses:**

1. Per-task variability and KV-reuse generality remains unexplored. While accuracy averages outperforms other pruning baselines, several benchmarks show worse performance (WNLI, SST-2, RTE), and the first/last-token KV-reuse rule lacks per-task diagnostics for mid-sequence evidence tasks.

2. The structural constraint may limit search space. The decode removal set must superset the prefill set to guarantee KV reuse. The trade-off versus potentially better non-subset solutions is not quantified.

**Questions:**

1. What guides 0.95 as the default of the distillation threshold, and how does it transfer across models?

2. Could you provide more task-specific analysis or adaptive policies would clarify when bandwidth cuts are safest other than first/last-token KV-reuse rule?

3. As the paper uses 256 examples from PIQA and MMLU as the default calibration set. How sensitive are removal decisions and final accuracy to the domain of the calibration data when evaluating on tasks like SST-2, WNLI, and RTE?

4. Bandwidth savings are now measured on a local network. Is it possible to consider application under different link speeds and latencies, including WAN-like RTTs, and with batch sizes beyond 1?

5. The method requires the prefill pruning set to be a subset of the decode pruning set to enable KV reuse. Could you quantify accuracy, latency, and bandwidth if this constraint is relaxed? If infeasible, explain why and if there can be feasible variants?

---

> ### Author Response · Authors · 2025-11-18
>
> Thanks for your detailed review and constructive feedback on our manuscript. We have carefully reviewed all your comments and addressed each concern with clear clarifications and supporting evidence. Please see our detailed responses below.
>
> **R1. Performance Analysis (W1).**
>
> Notably, none of the existing pruning methods is able to achieve the best performance across all benchmarks. In contrast, our approach consistently delivers substantially higher average performance in extensive large-scale evaluations.
>
> The fact that mid-sequence tokens often have low attention scores in the KV cache has already been demonstrated in several works, and is consistently observed across a wide range of tasks [1]. These intermediate tokens typically receive near-zero attention and can thus be safely ignored [1]. In essence, during the prefill stage, the attention mechanism enables the KV cache produced by later tokens to partially retain the information carried by the intermediate tokens. Compared with these methods, our approach introduces a lower load. Experiments in **Appendices J.1 and G** further show that our KV-cache pruning is both effective and robust across a wide range of task types, including those with very long intermediate sequences.
>
> [1] H\_2O: Heavy-Hitter Oracle for Efficient Generative Inference of Large Language Models
>
> **R2. Removal scheme (W2 and Q5).**
>
> We discuss this point in ​**Appendix M**​. If the pruning for prefill and decode does not follow a subset relationship, there are two major issues. First, it requires performing pruning independently for both stages, increasing the computational load. **More importantly, ​**without a subset relationship, a block retained in the decode stage may not exist in the prefill stage, making it impossible to transfer the corresponding KV Cache to the decode node and hindering execution.
>
> **R3. Distillation Threshold (Q1).**
>
> The cosine similarity of redundant blocks across different models (e.g., LLaMA and Qwen) is largely similar, mostly ranging from 0.92 to 0.97, and there is no significant difference in threshold settings between models. The value 0.95 is merely a hyperparameter choice, and as shown in ​**Appendix F**​, using other values can also achieve good performance.
>
> **R4. More Considerable Strategies (Q2).**
>
> We could dynamically adjust the front-to-back ratio parameters for specific tasks and reuse KV caches from previous steps that were not transmitted during execution. However, these approaches introduce additional overhead, which conflicts with our design objectives. As shown in Appendices J.1, J.2, and G, our current design performs well in terms of both load and accuracy.
>
> **R5. Robustness on Calibration Sets (Q3).**
>
> In ​**Appendix J.1**​, we show that consistent removal schemes**​ ​**can be achieved across different calibration sets.
>
> **R6. Data transmissions (Q4).**
>
> The essence of our method is to reduce the amount of data transferred from prefill nodes to decode nodes. As the batch size increases, the absolute reduction in data transfer also grows, while the proportion of data reduction remains roughly constant. This reduction ratio is maintained across different link speeds as well.

---

### Note · Authors · 2025-12-15

I have read and agree with the venue's withdrawal policy on behalf of myself and my co-authors.